# How does representation impact in-context learning: An exploration on a synthetic task

## Abstract

In-context learning, i.e., learning from in-context samples, is an impressive ability of Transformer. However, the exact mechanism behind this learning process remains unclear. In this study, we aim to explore this aspect from a relatively less explored perspective, i.e., representation learning. For in-context learning, the representation becomes more complex as it can be influenced by both model weights and context samples. To study how the model weights and in-context samples affect the prediction, we conceptually isolate the component, that can only be influenced by the model's weights, from the model's inner representation. We name this component as in-weights component and the rest as in-context component. We create a novel synthetic experimental set up, which allows to control the difficulty level of learning good in-context component, making it possible to study how the two components interplay with each other and impact the in-context performance. We find that the in-weights component plays a significant role in the learning of the in-context component. However, in traditional training way, the the in-weights component may be overlooked, resulting in a poor performance. We propose to a training setup to synergistically learn the in-weight and in-context components and the in-context learning performance can be significant improved. A further theoretical analysis is provided to justify the importance of our findings. Overall, our discoveries from the perspective of representation learning provide valuable insights into new approaches for enhancing in-context capacity.

## 1 Introduction

Transformer-based models have exhibited remarkable capabilities in language processing (OpenAI, 2023; Devlin et al., 2018). One of the most striking features is their ability to rapidly learn from contextual examples (Brown et al., 2020), which is referred to as in-context learning. As it does not require changing weights, in-context learning has garnered significant research interest and has been effectively employed to address real-world problems. This development calls for a deeper comprehension of the underlying mechanisms driving in-context learning. Numerous efforts have been dedicated to this crucial subject. For instance, several recent studies (von Oswald et al., 2022; Dai et al., 2023; Akyürek et al., 2022) have characterized in-context learning as a form of gradient descent. Other works (Li et al., 2023; Bai et al., 2023) have further interpreted in-context learning as algorithm implementation and selection.

In-context learning is different from regular supervised learning. For regular supervised learning, given the input sample $\mathbf{x}_p$ with label $y_p$, we want to find a parameterized function $f_\mathbf{w}$ with $\mathbf{w}$, such that $y_p = f_\mathbf{w}(\mathbf{x}_p)$. In this situation, all the information to accomplish the tasks is stored in the weights $\mathbf{w}$. However, for the in-context learning framework, there is extra information source, the context samples $s_c$. Then, the prediction can be modeled as $f_{\mathbf{w}, s_c}(\mathbf{x}_p)$, that means both the weights and the context samples can influence the prediction.

We study the joint effect between the in-weights component and in-context component on the prediction, by assuming that the function $f_{\mathbf{w}, s_c}(x)$ can be decomposited as $f_{\mathbf{w}, s_c}(x_p) = g(g_{weights}(\mathbf{x}_p), g_{context}(s_c))$. In this way, we can isolate a component $g_{weights}(\mathbf{x}_p)$ that only depends on the weights and the other component $g_{context}(s_c)$ that both depend on weights and context samples. We denote $g_{weights}(\mathbf{x}_p)$ as in-weights component and $g_{context}(s_c)$ as in-context

component. Due to the weights sharing within the transformer, $g_{weights}$ and $g_{context}$ interplay with each other.

## 1.1 MAIN CONCLUSIONS

With the aforementioned setup, we investigate the impact of the in-weight component and in-context component on in-context learning capabilities. The experimental results reveal that: $(i)$ In-weights components is easy to be overlooked in regular I.I.D training, leading to a poor in-context performance as the solid line shown in Fig. 6, even though it play an essential role in the in-context learning. Details is presented in Section 4. $(ii)$ By synergistically enhancing the in-weight and in-context components, the in-context performance of a base model (blue line in Fig. 6) significantly improves (in-context learning score increases from 0.168 to 0.885), and in-context ability emerges for small model (orange line in Fig 6).

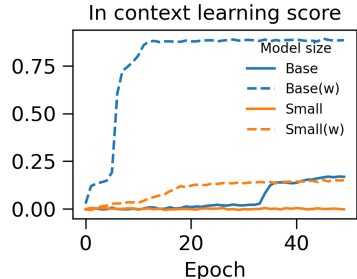

Figure 1: Test results for different training epoch and model size. Improving in-weights component (dashed line) can enable the emergence of in-context learning for small models. Detail in Section 4.2

A theoretical analysis is provided to further understand the role of in-weights component. We prove by construction that three additional Transformer layers on top of the representation with perfect in-weights component can achieve comparable performance in the experimental part.

## 1.2 CONTRIBUTIONS

Our main contributions can be summarized as follows:

- We give a formulation of in-weights and in-context component and introduce a new synthetic task that enables the study of the impact of *in-context component* and *in-weights component* on in-context ability.

- The experimental results reveal that although the in-weights component cannot directly influence the in-context learning performance, it plays a significant role in the learning of the in-context component. However, the traditional training method may overlook the in-weigts component and result in a worser performance.

- We offer mathematical analysis to understand the importance of in-weights components.

## 2 EXPERIMENTAL DESIGN

In this section, we will discuss the experimental design, which encompasses dataset construction, model and training objectives, and the exploration framework.

**Principles for Experimental Design** The following principles guide our experimental design: 1) The prediction of the prompt example should adapt to the in-context example. 2) The evolution of in-weights and in-context components should be trackable. 3) The learning of in-weights and in-context components must be controllable.

## 2.1 DATASET CONSTRUCTION

**In general tasks, the impacts of model weights and context samples are intertwined, which complicates the process of providing separate evaluations.** Therefore, we propose a task using the Shapes3D (Kim & Mnih, 2018) dataset for more controllable study. The experimental setting is shown in Fig. 2. Specifically, given a sequence of image and label pairs as context, the task involves predicting the label of the prompt image. Each image contains six different factors: object color, object shape, object scale, background color, floor color, and pose. We denote the factor as $e$ and the factor value of factor $e$ as $v^{(e)}$. For each sequence, we randomly choose a factor to generate the

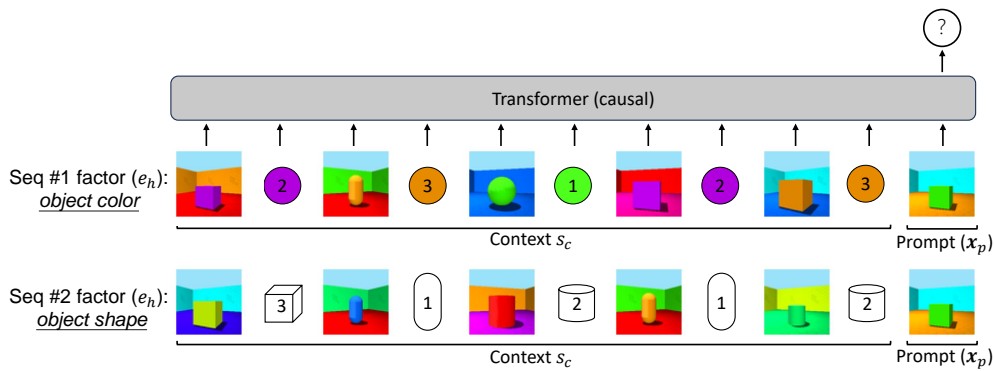

Figure 2: **Experimental setup**. We train Transformers using a sequence of image and label pairs, teaching the model to predict labels for each image. During inference, we evaluate the model's ability to accurately predict new, unseen sequences. The images from the 3D Shapes dataset are synthesized based on six factors. The output factor is determined by the context. In this case, we provide two sequences of factors: "object color" and "object shape," respectively.

labels of the images, referring to this factor as the **hidden factor** for this sequence. We denote the hidden factor fir $i$-th sentence as $e_i$. For the two context sequences in Fig. 2, the hidden factor of Seq #1 is object color, and the correct label for the prompt image is 1 (object color is green). In Seq #2, for the same prompt image, the correct label is 3 (object shape is cube).

In order to make accurate predictions, the network must not only identify the values of the six factors in the prompt image (associated with the in-weight component), but also identify the correct hidden factor to output based on the contexts (related to the in-context component).

## 2.2 PROBLEM DEFINITION

**Notations**  We denote $\mathbf{x}_p$ as the prompt example with ground truth label $y_p$ and the context examples are $s_c = \{\mathbf{x}_1, y_1, \cdots, \mathbf{x}_l, y_l\}$. We denote the factor values of prompt as $v_p$ and the corresponding factor value for factor $e$ as $v_p^{(e)}$. The hidden factor is denoted as $e_h$. We denote the mapping function as $m$, which maps the factor value to the corresponding label, i.e. $y_p = m(\mathbf{v}_p^{(e_h)})$. We denote the probability as $\mathbb{P}$.

**In-weights and in-context**  For regular supervised learning, given the input sample $\mathbf{x}_p$, we want to find a parameterized function $f_{\mathbf{w}}$ such that $y_p = f_{\mathbf{w}}(\mathbf{x}_p)$. In this situation, all the information to finish the tasks is stored in the weights $\mathbf{w}$. However, for the in-context learning framework, there is extra information source, the context samples $s_c$. Then, the predictions become $y_p = f_{\mathbf{w}, s_c}(\mathbf{x}_p)$, that means both the weights and the context samples can influence the prediction. For a representation $\mathbf{h}$ in the function $f_{\mathbf{w}, s_c}(\mathbf{x})$, we denote the component of $\mathbf{h}$ that can only be influenced by weights as in-weights component and the component that can be influenced by context examples as in-context component. The representation in $f_{\mathbf{w}}(\mathbf{x}_p)$ can be regarded as having only in-weights component. In the following, we decomposite the distribution $\mathbb{P}(y_p|\mathbf{x}_p, s_c)$ into the parts according whether they are depended on in-context examples.

**Proposition 2.1.** *The probability of $\mathbb{P}(y_p|\mathbf{x}_p, s_c)$ can be decomposite as:*

$$\mathbb{P}(y_p|\mathbf{x}_p, s_c) = \sum_{v_p, m, e_h} \underbrace{\mathbb{P}(y_p|v_p, m, e_h)}_{Properties\ of\ Task} \underbrace{\mathbb{P}(v_p|\mathbf{x}_p)}_{In\text{-}weights} \underbrace{\mathbb{P}(e_h|s_c, m)\mathbb{P}(m|s_c)}_{In\text{-}context}, \tag{1}$$

*where $\mathbb{P}(v_p|\mathbf{x}_p)$ is in-weights related information, and $\mathbb{P}(e_h|s_c, m)\mathbb{P}(m|s_c)$ is in-context related information. $\mathbb{P}(y_p|v_p, m, e_h)$ is related for the properties of task, and we have $\mathbb{P}(y_p|v_p, m, e_h) = 1$ if $m(v_p^{e_h}) = y_p$ else $\mathbb{P}(y_p|v_p, m, e_h) = 0$.*

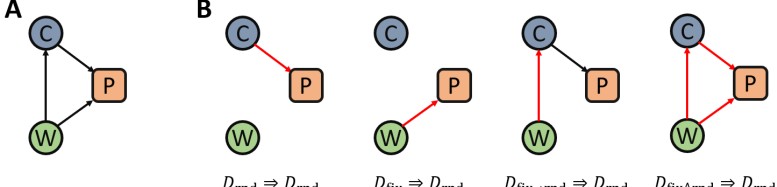

Figure 3: **Illustration of task design.** C, W, and P represent the in-context component, in-weights component, and in-context performance of the Transformer, respectively. A: All possible relations. B: the relationships we aim to investigate in each experimental setup. The red line signifies the relationship being explored, and the black line indicates the relationships that have not been removed.

Ideally, if we can approximate the distribution $\mathbb{P}(y_p|v_p, m, e_h)$ and the distribution $\mathbb{P}(e_h|s_c, m)\mathbb{P}(m|s_c)$, we can obtain the distribution $\mathbb{P}(y_p|\mathbf{x}_p, s_c)$. Based on the decomposited results of $\mathbb{P}(y_p|\mathbf{x}_p, s_c)$, we assume the representatio $\mathbf{h}$ can also be decomposited.

**Assumption 2.2.** *(Decomposible) We assume that there exsits functions $g, g_{weights}, g_{context}$, such that for any $(\mathbf{x}_p, s_c) \sim \mathbb{P}(\mathbf{x}_p, s_c)$, $f_{\mathbf{w}, s_c}(\mathbf{x}_p)$ can be decomposited as $f_{\mathbf{w}, s_c}(x_p) = g(g_{weights}(\mathbf{x}_p), g_{context}(s_c))$.*

*Remark* 2.3. **1) The function $g(g_{weights}(\mathbf{x}_p), g_{context}(s_c))$ is not the actually we calculate** $f(\mathbf{w}, s_c)$. It is a conceptually tool for us to decouple the influence of in-context samples' influence. **2) $g_{weights}$ and $g_{context}$ are only different in their inputs, which means that the $g_{weights}$ and $g_{context}$ may dependent with each other.** For example, in the Transformer $g_{context}$ may has the form $g_{context} = g'_{context}(\{g_{weights}(\mathbf{x}_1), y_1, \cdots, g_{weights}(\mathbf{x}_l), y_l\})$. This makes the in-weights and in-context components have very complex relation, which we aim to explore in this paper.

With the Assumption 2.2, we denote $g_{weights}(\mathbf{x}_p)$ as in-weights component and $g_{context}(s_c)$ as in-context component. Then, we define our expectation for the components to be good, that is to infer the corresponding part in Proposition 2.1.

**Definition 2.4.** *If $f_{\mathbf{w}, s_c}(\mathbf{x}_p)$ has good in-weights component in its representation, we can infer $\mathbb{P}(v_p|\mathbf{x}_p)$ from $g_{weights}(\mathbf{x}_p)$, and if it has good in-context component, we can infer $\mathbb{P}(e_h|s_c, m)$ from $g_{context}(s_c)$.*

### 2.3 EXPLORATION FRAMEWORK

To study whether we can have infer the can infer $\mathbb{P}(v_p|\mathbf{x}_p)$ from $\mathbf{h}_w$, and infer $\mathbb{P}(e_h|s_c, m)$ from $\mathbf{h}_c$, we leverage the probe method. and we defind the corresponding scores in the following

**Probing methods and metrics** We use three metrics here. **in-weights comp. score**: accuracy of the probe model to predict $v_p$ given $\mathbf{x}$, and "comp." is short for component. **in-context comp. score**: the accuracy of the probe model to predict $e_h$. **in-context learning score**: the gap between the accuracy of Transformer to predict $y_p$ give $l_1$ in-context examples and the accuracy of Transformer to predict $y_p$ give $l_2$ in-context samples. Here, we choose $l_1 = 40$ and $l_2 = 0$. The choose of $l_1, l_2$ doesn't have obvious influence (Olsson et al., 2022). **We give the detail of the probe framework design, probe model and training configure in the Appendix.**

To investigate the relationship between the effectiveness of learning in-weight/context components and the strength of in-context learning ability, we **control the difficulty of learning representation components by devising two label assignment settings** during the training phase:

- $D_{\text{fix}}$: The mapping $m$ remains constant throughout all sequences.
- $D_{\text{rnd}}$: The mapping $m$ is consistent within a sequence, but it is randomly choosed for different sequences.

The model is anticipated to learn the in-weight component more effectively in the $D_{\text{fix}}$ setting and the in-context component more effectively in the $D_{\text{rnd}}$ setting. To further analyze the interplay between these components, we consider two composite settings:

- $D_{\text{fix}\to\text{rnd}}$: In this setting, we initially train the model on $D_{\text{fix}}$ for a specific epoch, and then, we train the model on $D_{\text{rnd}}$.

- $D_{\text{fix}\wedge\text{rnd}}$: Half of the data in the training set utilizes the $D_{\text{fix}}$ setting, while the other half employs the $D_{\text{rnd}}$ setting.

Given two data settings $D_1$ and $D_2$, we use $D_1 \Rightarrow D_2$ to represent the evaluation result of a model on data setting $D_2$ after the model has been trained on data setting $D_1$.

**Analysis**   **For $D_{\text{fix}}$:** Recall that we have $\mathbb{P}(y_p|\mathbf{x}_p, s_c) \sim \mathbb{P}(v_p|\mathbf{x}_p)\mathbb{P}(e_h|s_c, m)\mathbb{P}(m|s_c)$. Under $D_{\text{fix}}$ setting, we only has one mapping function, which we denote as $m_0$. Therefore, we model can easily learn $\mathbb{P}_f(\mathbb{P}(m_0|s_c)) = \mathbb{P}(\mathbb{P}(m_0|s_c)) = 1$ and then the model only need to learn $\mathbb{P}(e_h|s_c, m)$ in its in-context component, which is expected to learn easier than $D_{\text{rnd}}$. And as a result, the model is expected to focus on learning $\mathbb{P}(v_p|\mathbf{x}_p)$. As a result, we are expected the model to have a better in-weights component in this setting. **Knowledge transfering between $D_{\text{fix}}$ and $D_{\text{rnd}}$ settings:** The knowledge of $\mathbb{P}(v_p|\mathbf{x}_p)$ is shared between these two tasks. The only different between these two tasks is $\mathbb{P}(e_h|s_c, m)\mathbb{P}(m|s_c)$.

**Task settings and explored relations (Fig. 3)**   Based on the analysis, Fig. 3 demonstrates the connections between the relations we wish to investigate (i.e., the relations between in-weight components, in-context components, and in-context learning) and the training-test data settings. The possible dependence between the in-weights and in-context components are caused by the possible dependent between $g_{weights}(\cdot)$ and $g_{context}(\cdot)$ as stated Assumption 2.2. For testing in-context ability during inference, we prefer the $D_{\text{rnd}}$ setting, referring to studies by Wei et al. (2023); Min et al. (2022), which indicate that the label-shuffled case can more effectively discern in-context ability.

## 3   COMPARED WITH PREVIOUS SYNTHETIC TASKS

Table 1: Comparison with other papers that explore in-context learning using synthetic dataset.

|  | Garg et al. (2022) | Chan et al. (2022a) | Ours |
|---|---|---|---|
| Synthetic task | Simple functions | Image data | Image data |
| Sentence Semantic | No | No | Yes |
| Perspective | Algorithm implementing | Data properties | Representation |
| In-weights learning | No | Trade off with in-context learning | Complex relations with in-context learning |

There are two kinds of synthetic tasks are common used in the exploration of in-context learning:

- (ST1, simple functions) In this task, a simple function is sampled for each sentence (an input sequence for the Transformer). Then, $x_i$ is generated by sampling from a specific distribution, and $y_i$ is produced using the sampled simple function with $x_i$ as input.

- (ST2, image sequence) In this task, $x_i$ is a randomly sampled image from the image dataset, and $y_i$ is generated using the original label values.

(ST1) is investigated in the works by Garg et al. (2022); von Oswald et al. (2022); Akyürek et al. (2022). (ST2) is examined in the studies by Chan et al. (2022a); Kirsch et al. (2022); Chan et al. (2022b). (ST1) researches in-context learning at a more abstract level, leading to the conclusion that in-context learning implements algorithms, such as gradient descent (von Oswald et al., 2022). However, their tasks are significantly distant from real applications because 1) The input token $x$ consists of numbers without any evident pattern or semantics, while most tokens in NLP tasks are words with clear meanings. 2) The tangible forms of their results is hard to be found in practice. The resolution of real NLP tasks is difficult to be expressed as straightforward, comprehensible algorithms, such as gradient descent or ridge regression. In contrast, (ST2) is closer to real applications since the image data used has semantic meaning. Thus, it is feasible to investigate how data properties influence in-context learning (Chan et al., 2022a). Our synthetic task belongs to the (ST2) category. The primary distinction between our synthetic task and the previous tasks in (ST2) is that "sentence semantics" are considered in our approach. In the following, we provide a detailed comparison between our work and that of Chan et al. (2022a).

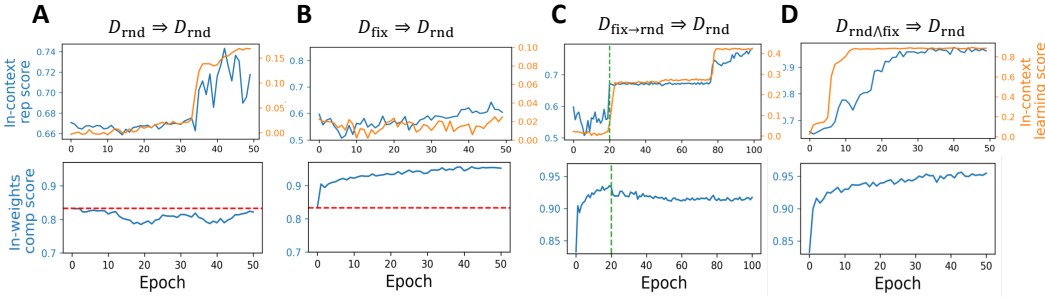

Figure 4: **Performance on $D_{\mathbf{rnd}}$ with different training settings. A**: $D_{\mathrm{rnd}} \Rightarrow D_{\mathrm{rnd}}$ only improve the in-context component. **B**: $D_{\mathrm{fix}} \Rightarrow D_{\mathrm{rnd}}$ only improve the in-weights component. **CD**: An improved in-weights component can speed up the learning process of the in-context component. The green dashed line marks the point when switch from $D_{\mathrm{fix}}$ to $D_{\mathrm{rnd}}$

**Comparison with Chan et al. (2022a)    Regarding dataset setting:** The primary difference in our task setting compared to that of Chan et al. (2022a) lies in the consideration of "sentence semantics." Specifically, if we remove hidden factors of the sequence, our constructed synthetic data would degenerate to that of Chan et al. (2022a). The **importance** of considering "Sentence Semantic" lies in the following: 1) Understanding sentence semantics plays a crucial role in practical applications, as evidenced by various studies (Zheng et al., 2021; Bowerman, 1976; Reimers & Gurevych, 2019). 2) Without sentence semantics, the function of in-context learning would degenerate into a simple copy-paste mechanism, wherein the Transformer can predict the label of a query image by searching for context images with the same label and then copying the label of the in-context image to the prediction of the query image. **Regarding the division of in-weights/in-context:** At a high level, the meanings of "in-context" and "in-weights" are consistent across our work and that of Chan et al. (2022a). However, our paper advances further by: 1) Analyzing the complex relationship between in-weights and in-context components from a representation perspective, leading to a more realistic conclusion. Chan et al. (2022a) concludes that in-context learning and in-weights learning are in a tradeoff relationship in their exploration, but large language models can exhibit both capacities Brown et al. (2020), which is acknowledged by Chan et al. (2022a) in the discussion at the end of their paper. 2) While Chan et al. (2022a) devises two tasks to evaluate in-context and in-weights learning, our work leverages a task that requires both in-context and in-weights information to solve. This setting is more closely aligned with practical applications, as real tasks often require both types of information (Brown et al., 2020; Alayrac et al., 2022).

## 4    RESULTS

In this section, we present our experimental results. In Section 4.1, we examine the in-weights component, in-context component and in-context learning performance score under different settings. In Section 4.2, we give further analysis of the influence of in-weights component under different settings. **We give the detail experiments setup, including model structure, optimization configure, training object in the Appendix.**

### 4.1    SEPARATE INFLUENCE OF IN-CONTEXT AND IN-WEIGHTS COMPONENTS

**Key Points**    Through experiments we observe that **1)** regular training $D_{\mathrm{rnd}} \Rightarrow D_{\mathrm{rnd}}$ will overlook the learning of in-weights component **2)** A high-quality in-weights component can assist in the learning of the in-context component, and **3)** Improving both the in-weights component and the in-context component simultaneously is more effective.

To verify the effectiveness of task design, we first conduct experiments to explore the evolution of representation learning when applying data settings $D_{\mathrm{fix}}$ and $D_{\mathrm{rnd}}$. We find that:

- **The regular training set up $D_{\mathbf{rnd}} \Rightarrow D_{\mathbf{rnd}}$ (the test and training data are from same distribution) cannot improve the in-weights component**. In Fig 4A, we observe the

**A**

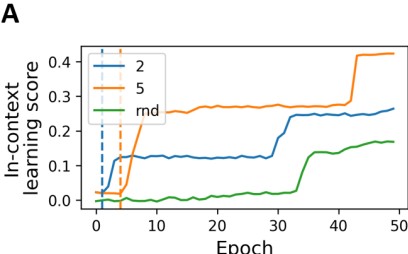

**B**

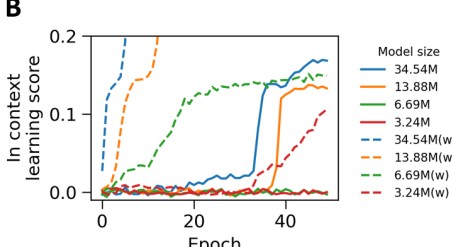

Figure 5: **Enhance in-weights and in-context components by combining $D_{\text{fix}}$ and $D_{\text{rnd}}$. A:** We conduct $D_{\text{fix}\rightarrow\text{rnd}} \Rightarrow D_{\text{rnd}}$ under different switching point. The curve with legend "2" means that we switch from $D_{\text{fix}}$ to $D_{\text{rnd}}$ at epoch 2. Curve with legend "rnd" is the baseline setting, i.e., $D_{\text{rnd}} \Rightarrow D_{\text{rnd}}$. The dash lines mark the corresponding switching points. **B:** Enhancing the in-weights component can facilitate the emergence of in-context learning ability in smaller models. The dashed line represents the task setting $D_{\text{fix}\wedge\text{rnd}} \Rightarrow D_{\text{rnd}}$, while the solid line corresponds to the setting $D_{\text{rnd}} \Rightarrow D_{\text{rnd}}$.

    increase of in-context comp. score and in-context learning performance. The in-weights comp. score stays below the initialized value.

- **A signicant improvement of in-weights learning under $D_{\text{fix}} \Rightarrow D_{\text{rnd}}$ settings** In Fig 4B, we observe a gradually improvement of in-weights comp. score. However, no obvious improvement of the in-context comp. score and in-context learning score can be observed.

We consider the setting $D_{\text{fix}\rightarrow\text{rnd}} \Rightarrow D_{\text{rnd}}$, where model is first trained on $D_{\text{fix}}$ to improve the in-weights component, and then we transfer to $D_{\text{rnd}}$ to enhance its in-context component. This approach allows us to examine the impact of the in-weights component on in-context components. We make the following observation:

- **A better in-weights component can accelerate the learning of the in-context component.** In Fig. 4C ($D_{\text{fix}\rightarrow\text{rnd}} \Rightarrow D_{\text{rnd}}$), we notice a sudden increase in the in-context comp score when switching from $D_{\text{fix}}$ to $D_{\text{rnd}}$. This result indicates that we can learn the in-context component more quickly based on a representation with a better in-weights component.

We then further investigate the combined effect of in-weights and in-context components on in-context learning by simultaneously improving the in-weights and in-context components using the task setting $D_{\text{fix}\wedge\text{rnd}} \Rightarrow D_{\text{fix}}$. We find that:

- **Learning in-weights component and in-context component simultaneously is more effective than learning them separately.** Compared to training on $D_{\text{fix}\Rightarrow\text{rnd}}$, the model trained on $D_{\text{fix}\wedge\text{rnd}}$ learns much faster( 4D ($D_{\text{fix}\wedge\text{rnd}} \Rightarrow D_{\text{rnd}}$)). Additionally, $D_{\text{fix}\wedge\text{rnd}}$ can facilitate in-context learning in smaller models (see Fig. **??**B).

## 4.2 INFLUENCE OF IN-WEIGHTS COMPONENT UNDER DIFFERENT SETTINGS

**Key Points** Through experiments we observe that **1)** Small epochs trained on $D_{\text{fix}}$ under $D_{\text{fix}\rightarrow\text{rnd}} \Rightarrow D_{\text{rnd}}$ setting can significant improve the in-context learning. **2)** Enhancing the in-weights component ($D_{\text{fix}\wedge\text{rnd}}$) can facilitate the emergence of in-context learning ability in smaller models.

We consider the setting $D_{\text{fix}\rightarrow\text{rnd}} \Rightarrow D_{\text{rnd}}$, where model is first trained on $D_{\text{fix}}$ to improve the in-weights component, and then we transfer to $D_{\text{rnd}}$ to enhance its in-context component. This approach allows us to examine the impact of the in-weights component on in-context components. We make the following observation:

- **More explorations on in-weights component** We conduct $D_{\text{fix}\rightarrow\text{rnd}} \Rightarrow D_{\text{rnd}}$ under different switching point. The results are given in Fig. 4A. We find that even training on $D_{\text{fix}}$ with small epochs, the model can still benifit a lot.

We then further investigate the combined effect of in-weights and in-context components on in-context learning by simultaneously improving the in-weights and in-context components using the task setting $D_{\text{fix}\wedge\text{rnd}} \Rightarrow D_{\text{fix}}$. We find that:

- **Enhancing the in-weights component can facilitate the emergence of in-context learning ability in smaller models.** In Fig. 4, we observe that enhancing the in-weights component by using $D_{\text{rnd}\wedge\text{fix}}$ settings, can facilitate the emergence of in-context learning ability in smaller models. For base models, we observe significant improvements of the in-context learning performance

## 5 FURTHER STUDIES OF THE IMPORTANCE OF IN-WEIGHTS COMPONENT

### 5.1 THEORETIC ANALYSIS

**Key points** We investigate the **mechanism** of in-context learning by construction. Our main result is that the significance of the in-weights component, as demonstrated in the experiments, is further validated by the fact that a simple Transformer can achieve potent in-context learning results based on the assumption of a perfect in-weights component.

We consider the naive Transformer (Vaswani et al., 2017). The hidden representation of token $i$ in Transformer is denoted as $\mathbf{h}_i \in \mathbb{R}^d$. The whole hidden state of the sequence is denoted as $\mathbf{H} = [\mathbf{h}_1, \cdots, \mathbf{h}_{2L}]^{\mathrm{T}} \in \mathbb{R}^{2L \times d}$. The hidden representation of $l$-th layer is denoted as $\mathbf{H}^{(l)}$.

**Definition 5.1.** *(Transformer) One layer of Transformer contains one attention layer and one MLP layer. The calculation of Attention Layer is*

$$\text{Attn}^{(l)}(\mathbf{H}^{(l)}) = \mathbf{H}^{(l)} + \sum_{c=1}^{C} \sigma\left(\mathbf{H}^{(l)}\mathbf{W}_Q^{(l,c)}(\mathbf{H}^{(l)}\mathbf{W}_K^{(l,c)})^{\mathrm{T}}\right)\mathbf{H}^{(l)}\mathbf{W}_V^{(l,c)}\mathbf{W}_O^{(l,c)}. \tag{2}$$

*And the calculation of MLP layer is*

$$\mathbf{H}^{(l+1)} = \text{Attn}^{(l)}(\mathbf{H}^{(l)}) + \text{Relu}(\text{Attn}^{(l)}(\mathbf{H}^{(l)})\mathbf{W}_1^{(l)})\mathbf{W}_2^{(l)}. \tag{3}$$

*Here we consider relaxed case where $\sigma = \text{Id}$.*

The relaxion of Transformer is discussed by many previous works. Press et al. (2019) discover using the Relu in feed forward layer can acheiver comparable results in original one. Wiegreffe & Pinter (2019); Brunner et al. (2019); Richter & Wattenhofer (2020) point out that softmax operation may not actually needed for Transformer.

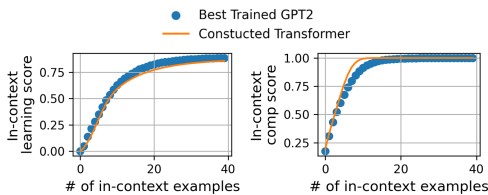

**Definition 5.2.** *(Perfect in-weights component) If feature $\mathbf{h}$ has a perfect in-weights component, then for all factor $e$, exists $\mathbf{W}_e \in \mathbb{R}^{d \times |V_e|}$ such that $\mathbf{f}_{\mathbf{x}_1}^{(e)} \cdot \mathbf{f}_{\mathbf{x}_2}^{(e)} = 1$ only when $v_{x_1}^{(e)} = v_{x_2}^{(e)}$, else we have $\mathbf{f}_{\mathbf{x}_1}^{(e)} \cdot \mathbf{f}_{\mathbf{x}_2}^{(e)} = 0$, where $\mathbf{f}_{\mathbf{x}}^{(e)} = \mathbf{W}_e h_{\mathbf{x}}$.*

Figure 6: The constructed Transformer can match the performancce of best trained GPT2 ($D_{\text{fix}\wedge\text{rnd}}$ setting) in experiment part

Based on the perfect in-weights component assumption, we can construct a Transformer with additional three layers to learn the in-context component and achieve comparable performance compared with the best trained GPT2 model in previous experiments.

**Proposition 5.3.** *We consider the data with $n_e$ factors and each factor has $n_v$ values in $D_{rnd}$ setting. For causal Transformer with the number of heads larger or equal the number of factors with the hidden size $\mathcal{O}(n_e n_v + L)$, if the Transformer can learn a perfect in-weights component in layer $k$, then it can learn a feature given $i$ in-context samples with in-context comp. score $\text{srs}_i = (1 - \text{srs}_{i-1})s_i + \text{srs}_{i-1}$ and $\text{srs}_0 = s_0$ at layer $k + 2$, where $s_i = 1 - \sum_{j=0}^{i} \binom{i}{j} \sum_{k=2}^{|E|} \binom{|E|}{k} \frac{k-1}{k} \left(\frac{(n_v-1)^{i-j}}{n_v^i}\right)^k \left(1 - \frac{(n_v-1)^{i-j}}{n_v^i}\right)^{|E|-k}$, and we can obtain in-context learning score as $\text{cls}_i = \frac{(n_v-1)(n_v^{i-1} - (n_v-1)^{i-1})}{n_v^i} \text{srs}_i$ at $k + 3$ layers.*

**In-context learning can be readily achieved with a high-quality in-weights component** Fig. **??** illustrates that the performance of the constructed simple Transformer can match the best-tuned GPT2 model under the perfect in-weights component assumption. Given the limited capacity of the three simple Transformer layers, it can be inferred that in-context learning can be easily achieved when supported by a high-quality in-weights component.

## 5.2 BEYOND SYNTHETIC TASKS

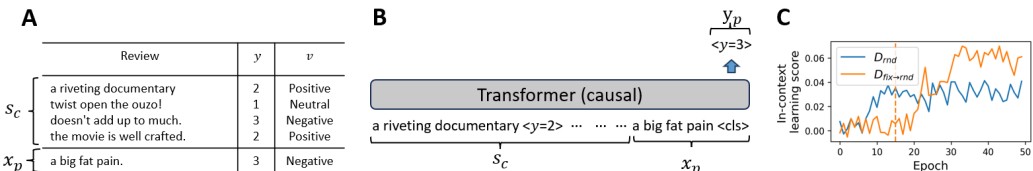

Figure 7: Experiments on SST-ICL dataset. **A:** Example of dataset. The example label $y$ is obtained from $v$ by map function $m$, i.e. $y = m(v)$. The in-context examples is denoted as $s_c$ and the prompt example is denoted as $x_p$. **B:** The Transformer is trained to predict the label of prompt example $y_p$ given $s_c$, $x_p$. **C:** Effect of in-weights component. We explore the $D_{\text{fix}\to\text{rnd}} \Rightarrow D_{\text{rnd}}$ and $D_{\text{rnd}} \Rightarrow D_{\text{rnd}}$ settings. The dash line denote the time when we transit from $D_{\text{fix}}$ to $D_{\text{rnd}}$.

We conduct another NLP meta learning task to demonstrate that the relations discovered in our synthetic task about in-weigts component, in-context component and in-context learning ability can further extend to practice problem.

**SST-ICL dataset** The dataset is contructed based on SST (Socher et al., 2013) datasets. We remove the long review in the datasets and transform the original labels into "Negative", "Positive" and "Neutral". Then, we organize the reviews follow the same way as that in Subsection A.4. We produce $10^4$ sequence for training and $4 \times 10^3$ for testing. Each sequence contains 5 reviews. We illustrate the example of the dataset in Fig. 9 AB.

**Experiments results** We follow the same training pipline as described in Section A. We compare the results of $D_{\text{fix}\to\text{rnd}} \Rightarrow D_{\text{rnd}}$ and $D_{\text{rnd}} \Rightarrow D_{\text{rnd}}$ settings. We find that improve of in-weights component (by training on $D_{\text{fix}}$) can also help the in-context learning in this setting.

## 6 CONCLUSION

This paper investigates the relationship between representation and in-context learning by decomposing representation into in-weights and in-context components. Our experiments demonstrate that the in-context component has a direct connection with in-context learning ability. Further investigation shows that the in-weights component plays a crucial role in the learning of the in-context component. These findings are further examined by constructing a simple Transformer that matches the performance of the best-trained GPT2 model. In summary, this paper unveils the influence of representation on in-context learning within the context of a synthetic dataset.

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

# A MORE DETAILS ABOUT EXPERIMENTS

## A.1 DATASET DETAIL

3dshapes[1] is a dataset of 3D shapes procedurally generated from 6 ground truth independent latent factors. These factors are floor colour, wall colour, object colour, scale, shape and orientation.

All possible combinations of these latents are present exactly once, generating N = 480000 total images.

Latent factor values floor hue: 10 values linearly spaced in [0, 1] wall hue: 10 values linearly spaced in [0, 1] object hue: 10 values linearly spaced in [0, 1] scale: 8 values linearly spaced in [0, 1] shape: 4 values in [0, 1, 2, 3] orientation: 15 values linearly spaced in [-30, 30] We varied one latent at a time (starting from orientation, then shape, etc), and sequentially stored the images in fixed order in the images array. The corresponding values of the factors are stored in the same order in the labels array.

## A.2 MODEL AND TRAINING CONFIGURE

In the proposed task, our objective is to investigate the properties of in-context learning. We utilize the causal Transformer, in which each token can only attend to prior tokens. Specifically, we implement the Transformer $f$ as the GPT2 model, consisting of 12 layers, 4 attention heads, and an embedding size of 128 in default. To simulate the auto-regression framework, we calculate the loss for the sequence $s = \{(\mathbf{x}_1, y_1), \ldots, (\mathbf{x}_L, y_L)\}$ as:

$$\mathcal{L}(\theta, s) = \frac{1}{L} \sum_{i=1}^{L} l(f_{\mathbf{w}, s^{(i-1)}}(\mathbf{x}_i), y_i), \tag{4}$$

where $s^{(j)} \triangleq \{\mathbf{x}_1, y_1, \cdots, \mathbf{x}_j, y_j\}$, $l$ denotes the loss function. $\mathbf{x}$ will be tokenized by VAE before being passed to Transformer. The training loss in the dataset $S$, which contains $n$ sequence, is calculated as the average of loss over all training sequences, i.e.,

$$\mathcal{L}(\theta, S) = \frac{1}{n} \sum_{s \in S} \mathcal{L}(\theta, s). \tag{5}$$

In this study, we employ the Adam optimizer (Kingma & Ba, 2014) and mini-batch training to optimize the loss function $\mathcal{L}(\theta, S)$. Here, we use cross-entropy as the loss function. We utilize a batch size of 128 and set the learning rate to 0.0001. For training purposes, we use $10^5$ sequences and, for evaluation, $4 \times 10^4$ sequences. There is no overlap between images in the training sequences and those in the evaluation sequences.

## A.3 MORE DETAIL ABOUT PROBE FRAMEWORK

We employ metrics for numerical evaluation of components and in-context learning performance. Since the components are hidded in the representation, we use the probe method (Alain & Bengio, 2016). The probe classifier has a single linear layer, with softmax and cross-entropy calculating the loss. It is trained until converge. The in-weight probe predicts values of six factors of all images, while the in-context probe identifies the hidden factor for each sequence. The details are as follows.

**In-context comp. score (Fig. 8A)** Given the dataset $S$, the in-context comp. score is calculated as $\frac{1}{|S|} \sum_{s \in S} \mathbf{1}_{\hat{e}_{h,s} = e_{h,s}}$, where $\mathbf{1}_{\text{expr}}$ is indicator function, $s$ is the sequence in the dataset, $e_{h,s}$ is the hidden factor for the sequence $s$, and $\hat{e}_{h,s}$ is the prediction of probe classifier. We use $|\cdot|$ to denote the corresponding size of a set.

**in-weights comp. score (Fig. 8B)** To remove the influence of in-context component, we disable the attention layer in the Transformer. We disable the attention layers by replacing the attention layers in the Transformer with identity maps, whose outputs are equal to their inputs. Then, the in-weights comp score is calculated as $\frac{1}{|S|} \sum_{s \in S} \frac{1}{|s||E|} \sum_{(x,y) \in s} \sum_{e \in E} \mathbf{1}_{\hat{v}_x^e = v_x^e}$, where $v_x^e$ is factor value of factor $e$ and sample $x$, $\hat{v}_k^e$ is the prediction of probe classifier, $s = \{(\mathbf{x}_1, y_1), \ldots, (\mathbf{x}_L, y_L)\}$ is the sequence in the dataset $S$ and $E$ is the set of all factors.

---

[1]https://github.com/deepmind/3d-shapes

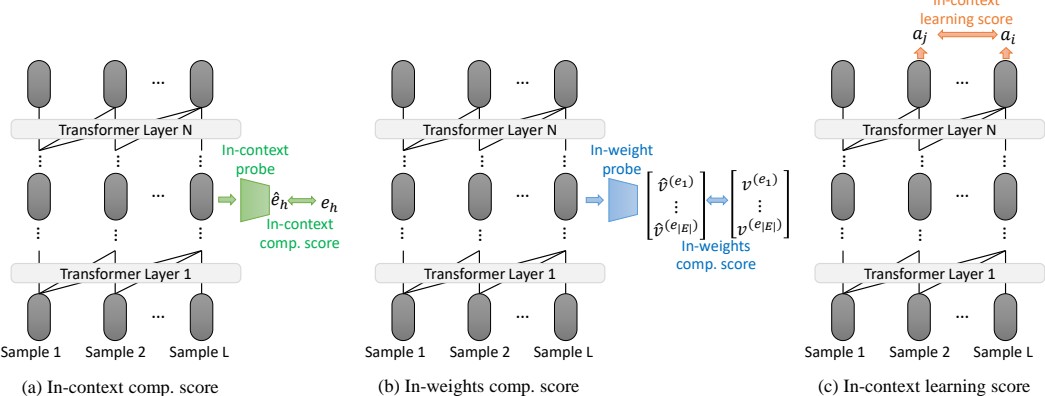

Figure 8: Illustration of calculation of in-context comp score(A), in-weights comp score(B) and in-context learning score(C).

**In-context learning score (Fig. 8C)** Let $a_i$ and $a_j$ be the accuracies of the Transformer on the $i$-th and $j$-th samples, respectively. Following Olsson et al. (2022), the in-context learning score is calculated as $a_i - a_j$. According to Olsson et al. (2022), the choice of $i, j$ on some reasonable range won't influence for the results. Here we choose $i = 40, j = 1$ by default. The reason is that under this setting, we have $a_i = (a_i - a_j) + a_j = (a_i - a_1) + a_1$. Because our task is unsolvable without in-context example, $a_1$ is a contant. In this case, the in-context learning score will have a same trend as the accuracy at $i$.

**Rationale to leverage hidden factor prediction as in-context probe task.** To solve the task on $D_{\text{rnd}}$, we need to obtain two informations from in-context, that is the hidden factors and the mapping between factor values and labels. Therefore, we can only choose the probe task for in-context component from these two candidates. **The mapping of the factor values and labels can not be used as probe task** because 1) the number all possible mapping is much larger than the size of dataset, which means we cannot learn the probe classifier. 2) The information of mapping is not neccessary stored in representation. Section 5.1 gives a solution of the model that no information of mapping stored in representation. **Hidden factors prediction is suitable for probe task** because 1) Hidden factor is the sequence level information that can only be learned from in-context example. 2) It is neccesary for solving the tasks and its information is stored in representation (Section 4.1).

### A.4 DATASET SPLIT

**In-context training** We first split all the the images in Shape3D into two part: the training image set (80 %) and the test image set (20 %). Then, we organize all the training images into $S_{\text{fix}}, S_{\text{rnd}}, S_{\text{fix}\wedge\text{rnd}}$, corresponding to $D_{\text{fix}}, D_{\text{rnd}}, D_{\text{fix}\wedge\text{rnd}}$ settings. $S_{\text{fix}}, S_{\text{rnd}}, S_{\text{fix}\wedge\text{rnd}}$ Test image set are also organized into $S'_{\text{fix}}, S'_{\text{rnd}}, S'_{\text{fix}\wedge\text{rnd}}$. Each of $S_{\text{fix}}, S_{\text{rnd}}$, and $S_{\text{fix}\wedge\text{rnd}}$ contains $10^5$ sequences. Each of $S'_{\text{fix}}, S'_{\text{rnd}}, S'_{\text{fix}\wedge\text{rnd}}$ contains $4 \times 10^4$ sequences.

**Probe model training** If we want to probe a model $f_{\mathbf{w},s_c}(\cdot)$ on setting $D_{\text{rnd}}$ (test setting), we will first train the probe model on $S_{\text{rnd}}$ with $f_{\mathbf{w},s_c}(\cdot)$ and we evaluate the probe model on $S'_{\text{rnd}}$ with $f_{\mathbf{w},s_c}(\cdot)$. The same for $D_{\text{fix}}$ and $D_{\text{fix}\wedge\text{rnd}}$ settings.

## B OTHER RELATED WORK

We discuss the most related work in the main part of paper. Here, we list other works that are weaker related to us.

**Analysis of Transformer** The analysis of Transformers can be broken down into two main components: examining the expressibility of Transformers and comprehending the mechanisms of learned Transformers. To analyze the expressibility of Transformers, a common approach is to determine

if they can solve specific problems by constructing appropriate weights. Giannou et al. (2023) demonstrates that Transformers can function as Turing machines, while Liu et al. (2022) shows that they can learn shortcuts to solve automata problems. In addition to expressibility, researchers have also investigated the mechanisms behind learned Transformers. Bietti et al. (2023) examines Transformers from a memory standpoint, and Tian et al. (2023) focuses on single-layer Transformers. While the analysis of Transformers is crucial to our work, our ultimate goal differs; we aim to bridge the gap between representation learning and in-context learning.

**Exploration of representation within Transformer.**    Owing to the widespread use of Transformers, numerous studies (Li et al., 2022; Voita & Titov, 2020) seek to investigate their internal representations as a means of comprehending their functionality. The most prevalent approach involves utilizing probe models and tasks to discern the information stored within these representations (Voita & Titov, 2020; Schouten et al., 2022). Taking a different perspective, Voita et al. (2019) explores the flow of information across Transformer layers and how this process is influenced by the selection of learning objectives. Our work shares similarities with these studies in that we employ the probe method to examine representations. However, our focus differs in that we do not concentrate on the semantic meaning within the representation. Instead, we investigate how the in-weights and in-context information impact representation.

## C   BRIDGE WITH PRACTICE

### C.1   RELATED EVIDENCE OF PRACTICE WORK REGARDING IN-WEIGHTS AND IN-CONTEXT COMPONENT

In this section, we provide evidence about that the in-context and in-weights components in practice tasks.

**Intuition 1: Influence of words replacing**    A key difference between the in-weights and in-context components lies in the susceptibility of the in-weights component to word substitution. The in-weights component can be easily disrupted if a word is replaced with a token that was not present during the training phase, as the weights lack information about this new token. On the other hand, if the context samples are rich in information, the meaning of this new token can still be deduced. This mirrors the human ability to infer the meaning of an unknown word based on its context. If word substitution leads to a decline in performance, it suggests that the Transformer's prediction relies heavily on the in-weights component.

**Intuition 2:Influence of number of in-context examples**    The efficiency of the in-context component is expected to rise with the inclusion of more context-specific examples, a characteristic not shared by the in-weights components, which remain unaffected by the addition of in-context examples. Therefore, if performance improves with the integration of more context-specific examples, it would suggest that the Transformer's prediction is heavily influenced by the in-context component.

**Intuition 3: Zero-shot performance**    The zero-shot performance can directly indicate the effectiveness of the in-weights component. This is because no in-context examples are provided in this scenario, reducing the problem to a traditional supervised one

Based on the intuitions above, we collect the related experiments in practice paper.

1. Min et al. (2022) discovered that (1) performance can be improved by increasing the number of in-context examples. (2) Changing the labels of in-context examples does not influence the predicted label. The first discovery indicates that the prediction relies on the in-context components. The second discovery suggests that the Transformer uses the in-weights component for label prediction, given that there is no observed change when the labels of in-context examples are altered.

2. Brown et al. (2020) found that larger models are increasingly effective at utilizing in-context information. This suggests that in real-world scenarios, the efficiency of the in-context component improves with the enlargement of the model's size. Brown et al. (2020) also found that enhancing the model size can boost its zero-shot capabilities. These findings suggest that scaling the model

can enhance both the in-weights and in-context components, and the model employs these two components to address the problem.

3.Wei et al. (2023) carried out research on a two-class classification issue. They conducted experiments in which they altered a certain percentage of labels in the context examples to ascertain if the model's prediction would also change. If a change was observed, it would imply that the prediction relies on the in-context components. If no change was noticed, the prediction would be considered to depend on the in-weights component. Their results were intermediate, suggesting that both in-weights and in-context components contribute. Additionally, they found that enhancing the model size increases the impact of in-context examples.

## C.2   BEYOND SYNTHETIC TASKS

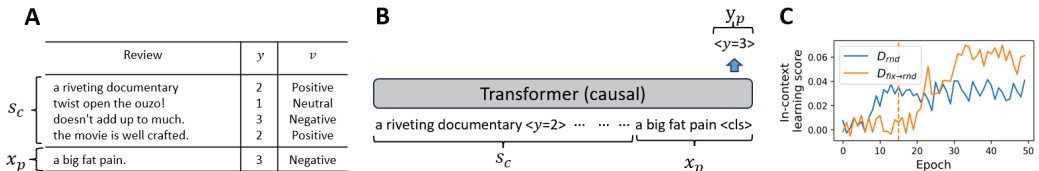

Figure 9: Experiments on SST-ICL dataset. **A:** Example of dataset. The example label $y$ is obtained from $v$ by map function $m$, i.e. $y = m(v)$. The in-context examples is denoted as $s_c$ and the prompt example is denoted as $x_p$. **B:** The Transformer is trained to predict the label of prompt example $y_p$ given $s_c$, $x_p$. **C:** Effect of in-weights component. We explore the $D_{\text{fix}\to\text{rnd}} \Rightarrow D_{\text{rnd}}$ and $D_{\text{rnd}} \Rightarrow D_{\text{rnd}}$ settings. The dash line denote the time when we transit from $D_{\text{fix}}$ to $D_{\text{rnd}}$.

We conduct another NLP meta learning task to demonstrate that the relations discovered in our synthetic task about in-weigts component, in-context component and in-context learning ability can further extend to practice problem.

**SST-ICL dataset**   The dataset is contructed based on SST (Socher et al., 2013) datasets. We remove the long review in the datasets and transform the original labels into "Negative", "Positive" and "Neutral". Then, we organize the reviews follow the same way as that in Subsection A.4. We produce $10^4$ sequence for training and $4 \times 10^3$ for testing. Each sequence contains 5 reviews. We illustrate the example of the dataset in Fig. 9 AB.

**Experiments results**   We follow the same training pipline as described in Section A. We compare the results of $D_{\text{fix}\to\text{rnd}} \Rightarrow D_{\text{rnd}}$ and $D_{\text{rnd}} \Rightarrow D_{\text{rnd}}$ settings. We find that improve of in-weights component (by training on $D_{\text{fix}}$) can also help the in-context learning in this setting.

## D   PROOF OF PROPOSITION 2.1

**Proposition D.1.** *Given $y_p$, probability of $\mathbb{P}(y_p|\mathbf{x}_p, s_c)$ can be decomposite as:*

$$\mathbb{P}(y_p|\mathbf{x}_p, s_c) = \sum_{v_p, m, e_h} \mathbb{P}(y_p|v_p, m, e_h)\mathbb{P}(v_p|\mathbf{x}_p)\mathbb{P}(e_h|s_c, m)\mathbb{P}(m|s_c). \tag{6}$$

*Proof.*

$$\begin{aligned}
\mathbb{P}(y_p|\mathbf{x}_p, s_c) &= \sum_{v_p, m, e_h} \mathbb{P}(y_p, v_p, m, e_h|\mathbf{x}_p, s_c) \\
&= \sum_{v_p, m, e_h} \mathbb{P}(y_p|\mathbf{x}_p, s_c, v_p, m, e_h)\mathbb{P}(v_p, m, e_h|\mathbf{x}_p, s_c,) \\
&= \sum_{v_p, m, e_h} \mathbb{P}(y_p|v_p, m, e_h)\mathbb{P}(v_p|\mathbf{x}_p, s_c, m, e_h)\mathbb{P}(m, e_h|\mathbf{x}_p, s_c) \\
&= \sum_{v_p, m, e_h} \mathbb{P}(y_p|v_p, m, e_h)\mathbb{P}(v_p|\mathbf{x}_p)\mathbb{P}(e_h|s_c, m)\mathbb{P}(m|s_c),
\end{aligned} \tag{7}$$

where the first equation is due to the law of total probability, the third equation is leverages the formular $\mathbb{P}(y_p|\mathbf{x}_p, s_c, v_p, m, e_h) = \mathbb{P}(y_p|v_p, m, e_h)$.

$\square$

## E   PROOF OF PROPOSITION 5.3

**Notation**   The position embedding is denoted as $\mathbf{p}_i = (0, \cdots 0, 1, 0, \cdots)$, where we only have value 1 at $i$-th position and 0 others. The weights for the attention operation of $l$-th layer and $c$-th head in Transformer is denoted as $\mathbf{W}_Q^{(l,c)}$, $\mathbf{W}_K^{(l,c)}$ and $\mathbf{W}_V^{(l,c)}$. The weights of forward layer in Transformer are denoted as $\mathbf{W}_1^l, \mathbf{W}_2^l$. We use $E$ to denote the all of all possible values of the factor $e$. we denote $\mathbf{y}_i$ as the one hot version of $y$. The vector with all zero values are denoted as $\mathbf{0} \triangleq (0, \cdots, 0)$.

We rewrite the proposition here.

**Proposition E.1.** *We consider the data with $n_e$ factors and each factor has $n_v$ values in $D_{rnd}$ setting. For causal Transformer with the number of heads larger or equal the number of factors with the hidden size $\mathcal{O}(n_e n_v + L)$, if the Transformer can learn a perfect in-weights component in layer $k$, then it can learn a feature given $i$ in-context samples with in-context component score as* $\mathrm{srs}_i = (1 - \mathrm{srs}_{i-1})s_i + \mathrm{srs}_{i-1}$ *and* $\mathrm{srs}_0 = s_0$ *at layer $k + 2$, where*
$s_i = 1 - \sum_{j=0}^i \binom{i}{j} \sum_{k=2}^{|E|} \binom{|E|}{k} \frac{k-1}{k} \left( \frac{(v-1)^{i-j}}{v^i} \right)^k \left( 1 - \frac{(v-1)^{i-j}}{v^i} \right)^{|E|-k}$, *and we can obtain in-context learning score as* $\mathrm{cls}_i = \frac{(n_v-1)(n_v^{i-1} - (n_v-1)^{i-1})}{n_v^i} \mathrm{srs}_i$ *at $k + 3$ layers.*

### E.1   PROOF OF USEFUL LEMMA

**Lemma E.2.** *One attention head can implement copy and past behavior.*

*Proof.* According to the definition of $\mathbf{p}_i$, we have $\mathbf{p}_i \cdot \mathbf{p}_j = 0$ if $i \neq j$, otherwise, we have $\mathbf{p}_i \cdot \mathbf{p}_j = 1$. We denote

$$\mathbf{M} = \begin{bmatrix} 0 & 0 & \cdots & 0 & 1 \\ 1 & 0 & \cdots & 0 & 0 \\ 0 & 1 & \cdots & 0 & 0 \\ \vdots & \vdots & \ddots & \vdots & \vdots \\ 0 & 0 & \cdots & 1 & 0 \end{bmatrix}.$$

Then we have $\mathbf{p}_i \mathbf{M} = \mathbf{p}_{i-1}$. For $j > i$, we denote the value of $2j$-th token as $\mathbf{h}_{2j} = (\mathbf{0}, \mathbf{0}, \mathbf{0}, \mathbf{0}, \mathbf{0}, \mathbf{p}_j)$ and $2i$-th token as $\mathbf{h}_{2i} = (\mathbf{h}_i', \mathbf{0}, \mathbf{0}, \mathbf{0}, \mathbf{0}, \mathbf{p}_i)$. If we want to copy the value of $2i$-th token to the value of $2j$-th token, we can set the query matrix as $\mathbf{W}_Q = (\mathbf{0}, \mathbf{0}, \mathbf{0}, \mathbf{0}, \mathbf{0}, \mathbf{M}^{j-i})$, the key matrix as $\mathbf{W}_K = (\mathbf{0}, \mathbf{0}, \mathbf{0}, \mathbf{0}, \mathbf{I})$ and value matrix as $\mathbf{W}_V = (\mathbf{W}_V', \mathbf{0}, \mathbf{0}, \mathbf{0}, \mathbf{0}, \mathbf{0})$. Then we have

$$\mathbf{h}_j^\mathrm{T} \mathbf{W}_Q \cdot \mathbf{h}_a^\mathrm{T} \mathbf{W}_K = \mathbf{p}_i \cdot \mathbf{p}_a = \begin{cases} 1 & a \neq j \\ 0 & a = j \end{cases} \tag{8}$$

Therefore, the $j$-th token can only attend to $i$-th token. Then we have the value of $\mathbf{h}_j$ after attention as $\mathbf{h}_j^{attn} = ((\mathbf{h}_i')^\mathrm{T} \mathbf{W}_V, \mathbf{0}, \mathbf{0}, \mathbf{0}, \mathbf{0}, \mathbf{p}_j)$. By setting $\mathbf{W}_V$ as different value, we can copy different part information of $i$-th to $j$-th token. Then the lemma is held.   $\square$

**Lemma E.3.** *For the input $\mathbf{h} = (\mathbf{h}_1^\mathrm{T}, \mathbf{h}_2^\mathrm{T}, \mathbf{h}_3^\mathrm{T})^\mathrm{T}$, where $\mathbf{h}_i \in \mathbb{R}^{d_i}$ and $d_1 + d_2 + d_3 = d$, for all $MLP_s(\mathbf{h}) = \mathbf{W}_2' \mathrm{Relu}(\mathbf{W}_1' \mathbf{h}_2) : \mathbb{R}^{d_2} \to \mathbb{R}^{d_2}$, there exists $MLP(h) = \mathbf{W}_2 \mathrm{Relu}(\mathbf{W}_1 \mathbf{h}) : \mathbb{R}^d \to \mathbb{R}^d$, such that $M\tilde{L}P(\mathbf{h}) = (\mathbf{h}_1, MLP_s(\mathbf{h}_2), \mathbf{h}_3)$.*

*Proof.* Obviously, for any $\mathbf{W}_1'$, there exists $\mathbf{W}_1$, such that $\mathbf{h}^{(1)} \triangleq \mathbf{h} \mathbf{W}_1 = (\mathbf{h}_1^\mathrm{T}, -\mathbf{h}_1^\mathrm{T}, (\mathbf{W}_1' \mathbf{h}_2)^\mathrm{T}, \mathbf{h}_3^\mathrm{T}, -\mathbf{h}_3^\mathrm{T})$.

Obviously, for any $\mathbf{W}_2'$, There exists $\mathbf{W}_2$, such that $\mathbf{h}^{(2)} = \mathbf{W}_2 \mathrm{Relu}(\mathbf{h}^{(1)}) = ((\mathrm{Relu}(\mathbf{h}_1) + \mathrm{Relu}(-\mathbf{h}_1))^\mathrm{T}, (\mathbf{W}_2' \mathrm{Relu}(\mathbf{W}_1'))^\mathrm{T}, (\mathrm{Relu}(\mathbf{h}_3) + \mathrm{Relu}(-\mathbf{h}_3))^\mathrm{T}) = (\mathbf{h}_1^\mathrm{T}, MLP_s(\mathbf{h}_2)^\mathrm{T}, \mathbf{h}_3^\mathrm{T})$   $\square$

### E.2   CONSTRUCTION OF TRANSFORMER

Without loss of generality, we assume the representation of Transformer in layer $k$ is in a form that $\mathbf{h}_{2i-1}^{(k)} = (\mathbf{f}_i, \mathbf{0}, \mathbf{0}, \mathbf{0}, \mathbf{0}, \mathbf{0}, \mathbf{0}, \mathbf{p}_i)^{\mathrm{T}}$ and $\mathbf{h}_{2i}^{(k)} = (\mathbf{0}, \mathbf{y}_i, \mathbf{0}, \mathbf{0}, \mathbf{0}, \mathbf{0}, \mathbf{0}, \mathbf{p}_i)^{\mathrm{T}}$ (Remind that one sample will takes two token, one for $x$ and one for $y$). Because the representation usually lays in low dimension space, a simple linear layer can transfer the representation in our defined sparse form. What's more, it is nature to assume that the position information is stored in the representation, since it is given in the input and it is essential for attention.

**Proof Sketch**   The constuction of weights can be divided into two steps: **1. Estimate the factor in this sequence.** According to the perfect in-weights component assumption, we can project the token feature into the space $\mathbf{f}^{(e)}$. The factor is chosed by find $e$ such that $(\mathbf{f}_i^{(e)})^{\mathrm{T}} \mathbf{f}_j^{(e)}$ can match $\mathbf{y}_i^{\mathrm{T}} \mathbf{y}_j$. (Layer 1) **2. Estimate** $y$. Based on the discovered factor in previous step, we 1) block the unrelated representation information (Layer 2) and 2) obtain the logit of new sample by comparing the similarity between the unblocked feature of this sample and the in-context sample. (Layer 3).

The consider the operations of Transformer in different layers.

**1) \*\* Layer 1 \*\***

Because we assume that $\mathbf{h}_{2i-1}^{(k)}$ is a perfect token representation, then there exists $\mathbf{W}_e$, such that $\mathbf{h}_{2i-1}^{(k)} \mathbf{W}_e = \mathbf{f}_k^e$, where $\mathbf{f}_i^{(e)}$ satisfies that $\forall e, i$, we have $\mathbf{f}_j^{(e)} \cdot \mathbf{f}_i^{(e)} = 1$ only when $v_i^{(e)} = v_j^{(e)}$ else $\mathbf{f}_j^{(e)} \cdot \mathbf{f}_i^{(e)} = 0$.

**Step 1, we use each attention head to obtain the matching information of each factor.**

We first consider the query token at the position $2i-1$ And we assign $\mathbf{W}_Q^{(l,k)} = \mathbf{W}_K^{(l,k)} = \mathbf{W}_{e_k}$ and $\mathbf{W}_V^{(l,k)} = (\mathbf{0}, \mathbf{0}, \mathbf{0}, \mathbf{0}, \mathbf{0}, \mathbf{0}, \mathbf{0}, \mathbf{I})^{\mathrm{T}}$ so that $(\mathbf{h}_i^{(l)})^{\mathrm{T}} \mathbf{W}_V^{(l,k)} = \mathbf{p}_i$.

$$\mathbf{b}_{e_k} = \sum_{a=1}^{2i-2} (\mathbf{h}_i^{\mathrm{T}} \mathbf{W}_Q^{(l,k)} \cdot \mathbf{h}_a^{\mathrm{T}} \mathbf{W}_K^{(l,k)}) \mathbf{h}_i^{\mathrm{T}} \mathbf{W}_V^{(l,k)} = \sum_{a=1}^{i-1} \mathbf{p}_a \mathbf{1}(v_a^{e_k} = v_i^{e_k}) \tag{9}$$

We denote base $= (2^0, 2^1, \cdots, 2^L)^{\mathrm{T}}$ and $\mathbf{u}_{2i-1} = (\{\text{base} \cdot \mathbf{b}_{e_1}, \cdots, \text{base} \cdot \mathbf{b}_{e_{n_e}}\}$. Obvious, there is $\mathbf{W}_O^{(l,k)}$ such that $\sum_{k=1}^{n_e} \mathbf{b}_{e_k} \mathbf{W}_O^{(l,k)} = (\mathbf{0}, \mathbf{0}, \mathbf{u}_{2i-1}, \mathbf{0}, \mathbf{0}, \mathbf{0}, \mathbf{0})$.

Then, we consider the token at position $2i$ as query token. We assign $\mathbf{W}_Q^{(l,n_e+1)} = \mathbf{W}_K^{(l,n_e+1)} = (\mathbf{0}, \mathbf{I}, \mathbf{0}, \mathbf{0}, \mathbf{0}, \mathbf{0}, \mathbf{0}, \mathbf{0})^{\mathrm{T}}$ and $\mathbf{W}_V^{(l,n_e+1)} = (\mathbf{0}, \mathbf{0}, \mathbf{0}, \mathbf{0}, \mathbf{0}, \mathbf{0}, \mathbf{0}, \mathbf{I})^{\mathrm{T}}$.

$$\mathbf{b}_y = \sum_{a=1}^{2i-1} (\mathbf{h}_i^{\mathrm{T}} \mathbf{W}_Q^{(l,n_e+1)} \cdot \mathbf{h}_a^{\mathrm{T}} \mathbf{W}_K^{(l,n_e+1)}) \mathbf{h}_i^{\mathrm{T}} \mathbf{W}_V^{(l,n_e+1)} = \sum_{a=1}^{i-1} \mathbf{p}_a \mathbf{1}(\mathbf{y}_a = \mathbf{y}_i) \tag{10}$$

Obvious, there is $\mathbf{W}_O$ such that $\mathbf{b}_y \mathbf{W}_O^{(l,n_e+1)} = (\mathbf{0}, \mathbf{0}, \mathbf{0}, \mathbf{u}_{2i}, \mathbf{0}, \mathbf{0}, \mathbf{0}, \mathbf{0})$, where $\mathbf{u}_{2i} = \{\text{base} \cdot \mathbf{b}_y, \cdots, \text{base} \cdot \mathbf{b}_y\}$. Note that base $\cdot \mathbf{b}_{e_k}$ has the property that base $\cdot \mathbf{b}_{e_k} = $ base $\cdot \mathbf{b}_{e_{k'}}$ if and only if all the context samples that have same factor value of factor $e_k$ with the sample $i$ also has the same factor value of factor $e_{k'}$ as sample $i$. **Therefore, we denote u as the matching information.**

After the operation, we have $\mathbf{h}_{2i-1} = (\mathbf{f}_i, \mathbf{0}, \mathbf{u}_{2i-1}, \mathbf{0}, \mathbf{0}, \mathbf{0}, \mathbf{0}, \mathbf{p}_i)$ and $\mathbf{h}_{2i} = (\mathbf{0}, \mathbf{y}_i, \mathbf{0}, \mathbf{u}_{2i}, \mathbf{0}, \mathbf{0}, \mathbf{0}, \mathbf{p}_i)$

**Step 2: compare the $\mathbf{u}_{2i-1}$ and $\mathbf{u}_{2i}$ to infer possible hidden factor.**

For embedding of $\mathbf{h}_{2i}$, using the copy past of Lemma E.2, we can obtain $\mathbf{h}_{2i} = (\mathbf{0}, \mathbf{y}_i, \mathbf{0}, \mathbf{u}_{2i}, \mathbf{u}_{2i-1}, \mathbf{0}, \mathbf{0}, \mathbf{p}_i)$. (By setting the copy position as $p_i$ and therefore the operation will only influence $y$ token.) According to Lemma E.3, there exists $W_1^{(l)}, W_2^{(l)}$, such that we have $\mathbf{h}_{2i} = (\mathbf{0}, \mathbf{y}_i, \mathbf{0}, \mathbf{u}_{2i}, \mathbf{u}_{2i-1}, \mathbf{m}_{2i}, \mathbf{0}, \mathbf{p}_i)$, where $\mathbf{m}_{2i} = \text{Relu}(\mathbf{u}_{2i} - \mathbf{u}_{2i-1}) + \text{Relu}(\mathbf{u}_{2i-1} - \mathbf{u}_{2i})$. Recall that $\mathbf{h}_{2i-1} = (\mathbf{f}_i, \mathbf{0}, \mathbf{u}_{2i-1}, \mathbf{0}, \mathbf{0}, \mathbf{0}, \mathbf{0}, \mathbf{p}_i)$. because all the corresponding terms of $\mathbf{h}_{2i-1}$ are $\mathbf{0}$, this operation won't impact the value of it.

The $k$-th position in $\mathbf{m}_{2i}$ is equal to $0$ if the values of $k$-th position of $\mathbf{u}_{2i-1}$ and $\mathbf{u}_{2i}$ are equal.

After this operation, we have $\mathbf{h}_{2i} = (\mathbf{0}, \mathbf{y}_i, \mathbf{0}, \mathbf{u}_{2i}, \mathbf{u}_{2i-1}, \mathbf{m}_{2i}, \mathbf{0}, \mathbf{p}_i)$ and $\mathbf{h}_{2i-1} = (\mathbf{f}_i, \mathbf{0}, \mathbf{u}_{2i-1}, \mathbf{0}, \mathbf{0}, \mathbf{0}, \mathbf{0}, \mathbf{p}_i)$

**2) ** Layer 2 ****

**Blocking the information according to $\mathbf{m}$.**

**First attention head:** for $\mathbf{y}$ token, at position $2i$, we apply Lemma E.2 to copy $\mathbf{m}_{2i-2}$ from $\mathbf{h}_{2i-2}$ to $\mathbf{h}_{2i}$. Due to the weights sharing of attention, this yield a iterative effects. We denote $\mathbf{m}'_{2i-1} = 2\mathbf{m}'_{2i-3} + \mathbf{m}_{2i-2}$ and $\mathbf{m}'_{2i} = \mathbf{m}_{2i-1} + \mathbf{m}_{2i}$. Therefore, we have $\mathbf{h}_{2i} = (\mathbf{0}, \mathbf{y}_i, \mathbf{0}, \mathbf{u}_{2i}, \mathbf{u}_{2i-1}, \mathbf{m}'_{2i}, \mathbf{0}, \mathbf{p}_i)$. Because of weights sharing, we have $\mathbf{h}_{2i-1} = (\mathbf{f}_i, \mathbf{0}, \mathbf{u}_{2i-1}, \mathbf{0}, \mathbf{0}, \mathbf{m}'_{2i-1}, \mathbf{0}, \mathbf{p}_i)$.

**Second attention head:** In this layer, for $\mathbf{y}$ token, at position $2i$, we apply Lemma E.2 to copy $\mathbf{f}_i$ from $\mathbf{h}_{2i-1}$ (This operation only affects $\mathbf{y}$ tokens). We have $\mathbf{h}_{2i} = (\mathbf{f}_i, \mathbf{y}_i, \mathbf{0}, \mathbf{u}_{2i}, \mathbf{u}_{2i-1}, \mathbf{m}'_{2i}, \mathbf{0}, \mathbf{p}_i)$.

**MLP Layer:** We denote $\mathbf{f}'_{i,x} \triangleq (\mathbf{f}_i^{(e_j)} - M\mathbf{m}'_{2i-1}[1], \cdots, \mathbf{f}_i^{(e_j)} - M\mathbf{m}'_{2i-1}[n_e])$ and $\mathbf{f}'_{i,y} \triangleq (\mathbf{f}_i^{(e_j)} - M\mathbf{m}'_{2i}[1], \cdots, \mathbf{f}_i^{(e_j)} - M\mathbf{m}'_{2i}[n_e])$. $M$ is a large constant value. In $\mathbf{f}'_{i,x}$, we will block the information of $j$-th factor if $\mathbf{m}'_{2i-1}[j] > 0$. $\mathbf{m}'_{2i-1}[j] < 0$ if and only if $\forall\, k < i,\ \mathbf{m}_{2k}[j] = 0$. The same for $\mathbf{f}'_{i,y}$. In MLP, we calculate $\mathrm{Relu}(\mathbf{h}_{2i-1}^{\mathrm{T}}\mathbf{W}_1^{(l+2)})\mathbf{W}_1^{(l+2)} = (\mathbf{f}_i, \mathbf{f}'_i)\mathbf{W}_1^{(l+2)} = (\mathbf{f}'_i - \mathbf{f}_i, \mathbf{0}, \mathbf{0}, \mathbf{0}, \mathbf{0}, \mathbf{0}, \mathbf{0}, \mathbf{0})$. Then, we have $\mathbf{h}_{2i-1} = \mathrm{Relu}(\mathbf{h}_{2i-1}^{\mathrm{T}}\mathbf{W}_1^{(l+2)})\mathbf{W}_1^{(l+2)} + \mathbf{h}_{2i-1} = (\mathbf{f}'_{i,x}, \mathbf{0}, \mathbf{u}_{2i-1}, \mathbf{0}, \mathbf{m}'_{2i-1}, \mathbf{0}, \mathbf{0}, \mathbf{p}_i)$. And similar, we have $\mathbf{h}_{2i} = (\mathbf{f}'_{i,y}, \mathbf{y}_i, \mathbf{0}, \mathbf{u}_{2i}, \mathbf{u}_{2i-1}, \mathbf{m}'_{2i}, \mathbf{0}, \mathbf{p}_i)$.

**3) ** Layer 3 ****

This layer obtain the logit of new sample by comparing the similarity between the unblocked feature of this sample and the in-context sample.

Setting $\mathbf{W}_Q^{(l+3,1)} = \mathbf{W}_K^{(l+3,1)} = (\mathbf{I}, \mathbf{0}, \mathbf{0}, \mathbf{0}, \mathbf{0}, \mathbf{0}, \mathbf{0}, \mathbf{0})$, we have $\mathbf{h}_i^{\mathrm{T}}\mathbf{W}_Q^{(l+3,1)} = \mathbf{h}_i^{\mathrm{T}}\mathbf{W}_K^{(l+3,1)} = \mathbf{f}'_i$. Setting $\mathbf{W}_V^{(l+3,1)} = (\mathbf{0}, \mathbf{I}, \mathbf{0}, \mathbf{0}, \mathbf{0}, \mathbf{0}, \mathbf{0}, \mathbf{0})$ such that $\mathbf{h}_{2i}^{\mathrm{T}}\mathbf{W}_V^{(l+3,1)} = \mathbf{y}_i$ and $\mathbf{h}_{2i-1}^{\mathrm{T}}\mathbf{W}_V^{(l+3,1)} = 0$.

For position $2i - 1$, we have

$$\mathrm{Logit} = \sum_{a=1}^{2i-2}(\mathbf{h}_i^{\mathrm{T}}\mathbf{W}_Q^{(l+3,1)} \cdot \mathbf{h}_a^{\mathrm{T}}\mathbf{W}_K^{(l+3,1)})\mathbf{h}_i^{\mathrm{T}}\mathbf{W}_V^{(l+3,1)} = \sum_{a=1}^{i-1}(\mathbf{f}'_{i,x} \cdot \mathbf{f}'_{a,y})\mathbf{y}'_a. \tag{11}$$

Note that value $\mathbf{f}'_{i,x} \cdot \mathbf{f}'_{a,y}$ is equal to the number of unblocked factors (both unbloked) that have same value between $a$-th sample and $i$-th sample Obviously, there is a $W_O^{(l+3,1)}$ such that $\mathbf{h}_{2i-1} = (\mathbf{f}'_{i,x}, \mathbf{0}, \mathbf{u}_{2i-1}, \mathbf{0}, \mathbf{0}, \mathbf{0}, \mathbf{m}'_{2i-1}, \mathrm{Logit}, \mathbf{p}_i)$.

Finally, we output Logit using the prediction head.

### E.3  PERFORMANCE ANALYSIS

Here, we will analyze the sequence representation score and the in-context learning accuracy of our constructed model.

**1) **In-context comp. score****

**Propability for same factor value between in-context examples and prompt sample** The probability for a in-context example having same value of a factor as prompt sample is $\frac{1}{n_v}$ and the probability of having different values is $\frac{n_v-1}{n_v}$. Therefore, given $i$ samples, the probability for $j$ samples have a same value of a factor as prompt sample is $\binom{i}{j}\frac{(n_v-1)^{i-j}}{n_v^i}$.

**Probability for connot distinguish factors** Given $i$ in-context examples, we cannot distinguish $k$ factors to decide which one is the hidden factor if the $k$ factors satisfying that of $\forall\, e_1, e_2 \in E_k,\ (x, y) \in s_c$, we have $v_x^{(e_1)} = v_p^{(e_2)} \Leftrightarrow v_x^{(e_2)} = v_p^{(e_2)}$, where $E_k$ is the set of these $k$ factors, $s_c$ is in-context examples, and $v$ is factor value.

Given $i$ in-context examples, the probability for we cannot distinguish $k$ factors is

$$\binom{|E|}{k} \sum_{j=0}^{i} \binom{i}{j} \left( \frac{(n_v-1)^{i-j}}{n_v^i} \right)^k \left( 1 - \frac{(n_v-1)^{i-j}}{n_v^i} \right)^{|E|-k}. \tag{12}$$

**In-context comp. score**   When we cannot distinguish the hidden factor from $k$ factors, the probability to predict wrong results is $\frac{k-1}{k}$. Combining the results above, we obtain the error that

$$\sum_{k=2}^{|E|} \binom{|E|}{k} \sum_{j=0}^{i} \binom{i}{j} \frac{k-1}{k} \left( \frac{(n_v-1)^{i-j}}{n_v^i} \right)^k \left( 1 - \frac{(n_v-1)^{i-j}}{n_v^i} \right)^{|E|-k}. \tag{13}$$

The probability to give a right prediction is

$$s_i = 1 - \sum_{j=0}^{i} \binom{i}{j} \sum_{k=2}^{|E|} \binom{|E|}{k} \frac{k-1}{k} \left( \frac{(v-1)^{i-j}}{v^i} \right)^k \left( 1 - \frac{(v-1)^{i-j}}{v^i} \right)^{|E|-k}. \tag{14}$$

In the constructed Transformer, we will autogressively combining the results of the previous prediction (Corresponding to Layer 2), we have:

$$\mathrm{srs}_i = (1 - \mathrm{srs}_{i-1})s_i + \mathrm{srs}_{i-1},$$

where $\mathrm{srs}_0 = s_0$.

**2) \*\*In-context learning score\*\***

The copy-past mechanism is used to predict the answer of the prompt example (Corresponding to layer 3). For the copy-past mechanism, have a in-context example with same prediction result as the prompt example is neccesary. When we correct predict the hidden factor, the probility to predict correctly is $1 - (\frac{n_v-1}{n_v})^i$. When we predict a wrong hidden factor, the probility is $\frac{1}{n_v}$. Combine the two above, we obtain the accuracy

$$\left(1 - (\frac{n_v-1}{n_v})^i\right) \mathrm{srs}_i + \frac{1}{n_v}(1 - \mathrm{srs}_i).$$

Because when no in-context example is given, the accuracy is $\frac{1}{v}$. Therefore we obtain the in-context learning score

$$\mathrm{cls}_i = \frac{(n_v-1)(n_v^{i-1} - (n_v-1)^{i-1})}{n_v^i} \mathrm{srs}_i.$$

### E.4   Contribution of the proof

**Comparison with previous work in analyzing the in-context learning mechanism**   The **key contribution** of our analysis is to analyze how the in-context learning learn the "sentence semantic",i.e. hidden factor, from in-context samples. Previous works investigate the mechanism of in-context learning mainly focus on the pair-wise relation between the query tokens and in-context tokens, and they (Olsson et al., 2022) reveals that copy-past is a important mechanism for in-context learning. In this paper, we want to analyze how Transformer learn the "Sentence Semantic". In our construction, we find that model may rely on the comparison of information within "Sentence" (i.e., pattern matching).

## F   Extra Experiments Result

In this section, we give some extra experiments results as complementary to the results listed in main text.

**More explorations on in-weights component**   We conduct $D_{\text{fix}\to\text{rnd}} \Rightarrow D_{\text{rnd}}$ under different switching point. The results are given in Fig. 10A. We find that even training on $D_{\text{fix}}$ with small epochs, the model can still benifit a lot.

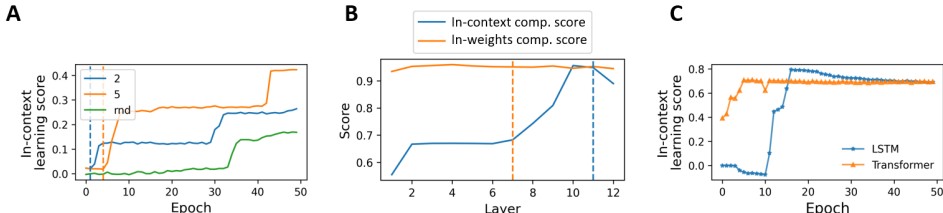

Figure 10: **A:** We conduct $D_{\text{fix}\to\text{rnd}} \Rightarrow D_{\text{rnd}}$ under different switching point. The curve with legend "2" means that we switch from $D_{\text{fix}}$ to $D_{\text{rnd}}$ at epoch 2. Curve with legend "rnd" is the baseline setting, i.e., $D_{\text{rnd}} \Rightarrow D_{\text{rnd}}$. The dash lines mark the corresponding switching points. **B:** The in-weights and in-context score when we probe at different layers. We choose the $D_{\text{fix}\wedge\text{rnd}} \Rightarrow D_{\text{rnd}}$ settings. The dashlines marked the chosen layers in the experiments. **C:** Comparison between the LSTM (Hochreiter & Schmidhuber, 1997) and Transformer.

**Abalation study on the probe layers** We give the ablation study on different probe layers. We choose the setting $D_{\text{rnd}\wedge\text{fix}} \Rightarrow D_{\text{rnd}}$ and the model trained at 50 epoch. The choose of setting is arbitrary. The in-weights and in-context comp. score has similar trend across layers for same model trained on different task settings and different epoch (except extremely close to initialization). The results are displayed in Fig. 10B.

**Compared between Transformer and LSTM.** In Section 4, our primary focus is on discussing the results of the Transformer. In this section, we aim to compare the in-context learning abilities of the Transformer and LSTM models. The LSTM model consists of 6 layers, while the Transformer has 4 heads, 6 layers. We examine the scenario where $D_{\text{fix}} \to D_{\text{fix}}$. Unfortunately, we are unable to train the LSTM model for all other cases. During the comparison, we employ a larger hidden size for the LSTM model, as it tends to fail when using a smaller hidden size. The results are displayed in Fig. 10C. We find that 1)LSTM is much harder for obtain in-context learning compared with Transformer. 2) LSTM has potential to obtain better in-context learning ability.

