# OpenReview forum: "How does representation impact in-context learning: An exploration on a synthetic task"
_ICLR.cc/2024/Conference — Submitted to ICLR 2024_

### Official Review · Reviewer_gm8Y · 2023-10-15

**Soundness:** 2 fair
**Presentation:** 2 fair
**Contribution:** 2 fair
**Rating:** 3
**Confidence:** 4

**Summary:**

This study investigates how Transformers learn from in-context examples, a process still not fully understood. It explores this phenomenon through representation learning, considering both model weights and in-context samples. The findings show a strong link between the quality of in-context representations and learning performance. A well-developed in-weights component enhances in-context learning, suggesting it should be foundational. Besides, a simple Transformer, using pattern matching and copy-paste for in-context learning, matches complex models assuming perfect in-weights. These insights offer new perspectives for enhancing in-context learning abilities.

**Strengths:**

- The research topic is realistic and important. In-context learning has attracted lots of attention from the research community. It is significant to understand its inherent mechanism for the development of more advanced techniques.
- The provided perspective that analyzes in-context learning is interesting and may inspire follow-up research.

**Weaknesses:**

- Technical contributions of this paper are limited.
- The writing should be polished further. For the current form, there is a series of unclear explanations, descriptions, and justifications.

More details and concerns can be checked in "Questions".

**Questions:**

- This paper claims that it attempts to understand in-context learning from representations. However, throughout this paper, there is no clear definition of the representation in in-context learning.
- How to understand "a good in-weight component" in Section 1.1. Could the paper provide more details? This is hard to understand from the current descriptions.
- I am confused about the approximation that uses a very simple network to estimate the large language models (LLMs). This is a bit incredible. If the approximation can be so good with respect to performance, why do we still increase the scale of LLMs? Would the approximation bring some disadvantages?
- Could the paper provide representative examples of the data of NLP not just images (c.f., Figure 2)?
- In this paper, $i=40$ and $j=0$. I am confused about the determination of such values.
- At the beginning of this paper, it claims PCA is employed. However, after checking the paper, the technical details and importance of PCA are not clearly stated.
- Could the paper supplement some intuitive explanations about Definition 2 for better understanding?
- The limitation of this paper (which is also stated in Section 4) is worrying. Perhaps, due to the issue, the obtained conclusion is not general.
- This paper looks like it was submitted in a hurry. There are many typos in the current form. For example,
(1) It should use upper and lower quotation marks but not all lower quotation marks.
(2) On the citation of the Adam optimizer.
(3) In-weights comp. but not in-weights comp.
(4) There should be a blank space between descent and (von Oswald et al., 2022). Similar, between Transformer and (Vaswani et al., 2017), between this sequence and According, and between semantic” and i.e..
Please check them carefully for better representations.

Due to the above concerns, before rebuttal, I am negative for the current form of this paper. I am glad to hear responses from authors and comments from other reviewers, and increase my score accordingly.

**Details Of Ethics Concerns:**

NA.

---

> ### Author Response · Authors · 2023-11-17
>
> Thank you for your valuable comments and suggestions and the encouraging words. We appreciate the time and effort you took to review our paper. Please find below our responses to your questions and the changes we have made to address them.
>
> Q1,Q2. Thank you for your pointing out. It is an important question. We rewrite the section 2.2 to solve these problems. In the revision of Section 2.2 , we give a formal definition about in-weights and in-context components, formal definition about what are good in-weights and in-context components and a decomposition of the problem.
>
> Q3, We regret that our writing makes you misunderstand here. We are not asserting that a small network can approximate the Large Language Model (LLM). Our main points are: 1) The in-weights component plays a crucial role in in-context learning. This is evident as, under the perfect in-weights component assumption, even a simple network can exhibit proficient in-context learning. 2) We demonstrate that the neural network we've constructed can approximate the performance of the model trained on experimental data, thereby validating that our construction is meaningful.
> We revise the paper to reduce the possible of the misunderstanding.
>
> Q4, Yes. We give a experiment on a new NLP dataset to demonstrate that conclusion obtained in synthetic tasks can be extended to NLP task in Appendix C.2.
>
> Q5, The selection is arbitrary. As stated in [1], the choice of i, j within a reasonable range will not affect the results.
>
> Q6, We regret that our writing makes you misunderstand here. Our intention in using PCA was to elucidate why we named them the "in-weights component" and "in-context component". However, we did not mean to imply that we are specifically using PCA. To avoid this ambiguity, we have revised the paper.
>
> Q7, This definition is to ensure that the representation has a good in-weights component. (the definition of ``good in-weights component" is given in section 2) and the in-weights component can be easily readout using a linear layer.
>
> Q8, We add the Section C in the Appendix. We give the following effort to try to solve your concern: 1) we analyze the previous discovery in NLP paper related to in-weights and in-context components. 2) we add a new experiment on a NLP dataset and our discoveries are held in this dataset.
>
> Q9, We will revise these typos.
>
> [1] Catherine Olsson, Nelson Elhage, Neel Nanda, Nicholas Joseph, Nova DasSarma, Tom Henighan, Ben Mann, Amanda Askell, Yuntao Bai, Anna Chen, et al. In-context learning and induction heads. arXiv preprint arXiv:2209.11895, 2022.

---

### Official Review · Reviewer_MkMt · 2023-10-31

**Soundness:** 2 fair
**Presentation:** 3 good
**Contribution:** 3 good
**Rating:** 5
**Confidence:** 3

**Summary:**

In this paper, the authors explain in-context learning in Transformers. They break down learning into two components: in-context and in-weight and introduce two probes to investigate these components. The in-context probe measures how well the model predicts the correct factor in a task sequence, while the in-weight probe assesses how well the model learns the overall task. To investigate this empirically, they use a synthetic dataset where an image of a 3D shape is given, and the model predicts one of six factors.  They evaluate the effectiveness of probes by look at its predictions on model trained on four different dataset variations. They find that (1) the in-weight component is effective with a fixed mapping between factor values and labels; (2) The in-context component is effective with randomly shuffled mappings, and in-context learning from random mappings generalizes to fixed mappings; (3) Switching from fixed to random mappings accelerates in-context learning; (4) Training on a mix of fixed and random mappings is more effective than training on each individually. Overall, the authors observe a strong correlation between the in-context component and  in-context learning performance and also that in-weight component influences in-context learning but is not sufficient. Furthermore, they show that a  constructed Transformer model with perfect in-weight component matches the performance of a trained GPT2 model, highlighting the importance of the in-weight component for in-context learning. Lastly, they suggest that two-mechanisms used by the constructed transformer - pattern matching and copy-paste - might be the mechanisms underlying in-context learning in larger models, such as Large Language Models (LLMs).

**Strengths:**

- the paper is addressing a relevant topic and ICLR is an appropriate for the submission
- the approach of probing in-context and in-weights representation is definitely an interesting and the authors take a look at it from representation learning perspective unlike previous who have looked into from data-distributional perspective (Chan et al. 2022) and casting it as gradient-based learning problem (von Oswald et al. 2022).
- details of task, the model and the experimental setup is quite clearly laid out. For most results, the hypotheses were mentioned along with the intuition before presenting the results which made following the results quite straight forward.
- although the organisation of the text is a bit unconventional, the authors managed to successfully get across the main points pretty well. apart from a few minor typos the figures also manage to convey the results quite well.

**Weaknesses:**

- given how the different experiments and probes are set-up, the results look a bit obvious to me.  For example, `D_fix` → `D_rand` results was quite trivial as during training time the model never saw such tasks. It would be interesting to understand within the same framework under what conditions does in-context learning does not emerge. Are there cases where in-weight learning interferes with in-context learning and vice-versa? For example, what if you overtrain on `D_fix`/ `D_rand` or under/over parameterise the model? what would your prediction be for this case?
- one of the strengths of the task is that it requires  the model to do both in-context and in-weight learning to do well. Have the authors investigated the phenomenon in purely language setting where this is also a requirement?

**Questions:**

- what is the loss function? is it `cross-entropy loss`?
- the proof by construction was a bit convoluted to understand, quite a few notations where missing or unclear. atm, I am unsure if I fully understand it. Perhaps this was because of the lack of space or written a bit at the end but I would be happy to go through it again if expanded with intuitions.


**Minor edits**

- Pg 4: Adam optimizer [cite: kingma2014adam]
- Pg 4: Since the components are hidden in the representation (under exploration framework section)
- Pg 3: `ei` is the hidden factor of the i-th sequence or i-th element in the sequence?
- Pg 4: `defailt`
- Pg 4 : consistency for capitalisation
- Figure 5: in-context rep score and in-context comp. score
- no abbreviation specified before using comp. for coponent
- Figure 5A: typo in the title D_fix→ rand
- Pg 8: typo `fead forward layer can achiever`

---

> ### Author Response · Authors · 2023-11-17
>
> Thank you for your valuable comments and suggestions. We appreciate your feedback and have made several changes to address your concerns. Please find below our responses to each point you raised.
>
> ### Weakness 1
>
>  D_fix can be considered a specific instance of D_rand. Even in the "D_rand->D_rand" scenario, the test set includes tasks that the model has not previously encountered. This is because the number of potential mappings between factor values and labels is $n_e(n_e-1)(n_e-2)...(1)$, where n_e represents the total possible factor values of factor e. If n_e equals 10, then the number of potential mappings, $n_e(n_e-1)(n_e-2)...(1)$, is approximately 3.6*10^7. This figure greatly exceeds the number of sequences (10^5) present in the training set.
>
> ### Weakness 2
> Thanks for your suggestion. We add this in Section C (Appendix). In this section, we summary some prior practice works that evident both in-weights and in-context components are used in practice. And we give an experiment on a new NLP dataset to demonstrate that conclusion obtained in synthetic tasks can be extended to NLP task.
>
> ### Question
>
> Q1: yes, it is crossentropy. Thank you for your pointing out. We add the description in the paper.
>
> Q2： We have entirely rewritten the proof to enhance its readability (Section E in Appendix). To accomplish this, we've taken the following steps: 1) We've added a new paragraph to explain the notation, and 2) we've incorporated more explanations and intuitive reasoning throughout the proof.
>
> ### Minor edits
>
> We fix the issues. We will continue to check potential grammar issues.

---

> > ### Comment · Reviewer_MkMt · 2023-11-21
> >
> > Although I did not ask for the text mentioned in Appendix C.1, it was a good addition in terms of providing more context as to why you are doing this. However, please note that there are several typos and grammar errors in this text.
> > I thank the authors for validating the probes on an NLP task and showing that at least some of the findings hold.
> > I also thank the authors for improving the proof by providing additional details and intuition. However, I feel that this proof, in its current state, is still difficult for me to understand. I do not feel confident enough to approve it yet, but perhaps one of the other reviewers can confirm if it is correct.
> > Additionally, when considering the comments left by the other reviewers, I see that most reviewers share my concern that the finding — that a good in-weight component is required for in-context learning — is more or less expected. Whether it holds for larger models that are strong in both aspects still seems to be a matter of debate.
> > Lastly, I believe that the writing needs further improvement as there are several grammar issues, including in the newly added text, and gaps in several explanations/notations. I agree with one of the reviewers that at least parts of the paper — if not all — still seem to be hastily written.
> > Therefore, taking all of these points into consideration, I have decided to keep my score as is at 5.

---

> ### Author Response · Authors · 2023-11-23
>
> **Point 1: please note that there are several typos and grammar errors in this text**: We will fix the grammar errors.
>
>
> **Point 2: I feel that this proof, in its current state, is still difficult for me to understand.**
>  Please respect our time for this rebuttal. There exist two elements that influence the comprehensibility of a proof: the complexity of the issue at hand and the quality of the proof's composition. If the former is the case, it cannot serve as a valid reason to dismiss our paper. However, if it's the latter, **please provide us with detailed feedback rather than a mere "However, I feel that this proof, in its current state, is still difficult for me to understand"**. This behavior is irresponsible.
>
> **Point 3: I see that most reviewers share my concern that the finding — that a good in-weight component is required for in-context learning — is more or less expected. Whether it holds for larger models that are strong in both aspects still seems to be a matter of debate.** The most reviewers concern about the experiments about D_fix->D_rnd and "in-context component is related to in-context learning" is within expectation, instead of the conclusion you mentioned that "that a good in-weight component is required for in-context learning". You also don't raise the concern in your first review. What's more, "that a good in-weight component is required for in-context learning — is more or less expected" and "Whether it holds for larger models that are strong in both aspects still seems to be a matter of debate" are somewhat contradicted with each other. If it is within expectation, then why you questions about  "Whether it holds for larger models that are strong in both aspects still seems to be a matter of debate"?

---

### Official Review · Reviewer_EdZW · 2023-11-01

**Soundness:** 3 good
**Presentation:** 3 good
**Contribution:** 3 good
**Rating:** 5
**Confidence:** 2

**Summary:**

This paper investigates the effects of in-weight and in-context components on In-Context Learning (ICL) capacity. The authors conduct a series of experiments demonstrating the significant role of the in-context component in the success of ICL. They also show that a well-trained in-weight component enhances the learning of a high-quality in-context component. Remarkably, the authors establish that, with an optimally pre-trained in-weight component, it is possible to construct additional Transformer layers (three in total) to effectively learn the in-context component. This yields performance on par with certain pre-trained Transformer models.

**Strengths:**

1.  The authors adopt an approach to evaluate the implicit in-context and in-weight components by employing probes. This is a significant departure from direct parameter analysis, which is challenging since in-weight and in-context parameters are tightly integrated. They develop a scoring system for the in-context and in-weight components, a novel and logical idea.

2. The paper is well-structured, and the authors provide numerous relevant experiments followed by comprehensive discussions.

3. The paper not only empirically demonstrates the impact of the in-context and in-weight components on ICL capacity but also provides theoretical evidence proving the existence of construction that can perform in-context learning using three additional Transformer layers, given perfectly pre-trained in-weight component. This claim is further validated through experimental evaluation.

**Weaknesses:**

1. There are areas in the manuscript that require attention, such as citation error in Section 2.2.

2. The data settings $D_\text{rnd}$ and $D_\text{fix}$ used in the experiments do not sufficiently capture the separate impacts of in-context and in-weight components.

    - $D_\text{rnd}$ seems to still take the in-weight component into consideration, given its nature as a statistical classification problem.
    - $D_\text{fix}$, on the other hand, seems to be more aligned with in-weight learning. However, it does not fully represent the in-weight component implicit in the original dataset. Given that in-context learning involves numerous classification tasks with shared inputs but varying labels, a more holistic approach considering all these tasks is necessary to capture the complete in-weight component.
3. The probe methodology requires further clarification. There is a lack of clarity regarding from which layer features are drawn and how the choice of layer affects the results. It is possible that some layers dominantly contain in-context components, while others are richer in in-weight components. Additionally, the rationale behind training each classifier for only one epoch, as opposed to ensuring thorough and adequate training, is not explained.

**Questions:**

1. Could the authors provide more details on the implementation of the probe methods, as mentioned in the Weaknesses section of this review?

2. In Section 2.3, the statement "we disable the attention layer in the Transformer" is made. Could the authors detailed on this?

---

> ### Author Response · Authors · 2023-11-17
>
> We appreciate the time and effort you took to review our paper. Please find below our responses to your concerns and the changes we have made to address them.
>
> ### Weakness 1
>
> Thank for your pointing out. We have fixed it in the paper.  We will continue to check these issues.
>
> ### weakness 2
> We regret that our writing makes you misunderstand here. The role of D_rnd and D_fix setting is not to seperate the impacts of in-context and in-weights component. Instead, we use the probe method to seperate it. In the revision, we rewrite the Section 2 to make it clear. In the revision, we provided a formal definition of the in-weights and in-context component and an analysis of the D_fix and D_rnd settings. We think the revision can solve your concern. (Section 2.2 and the paragraph start with "analysis" in section 2.3 are directly related to this).
>
> ### Weakness 3
>  **Point: There is a lack of clarity regarding from which layer features are drawn and how the choice of layer affects the results. It is possible that some layers dominantly contain in-context components, while others are richer in in-weight components.** We add the experiment in Section F (Appendix). In the experiment, we find that the in-context component concerntrate on the higher layer whether and in-weights component prefer lower layers. We also give the choosen layers.
>
> **Point:  the rationale behind training each classifier for only one epoch, as opposed to ensuring thorough and adequate training, is not explained.** In this paper, we use linear probe, (The probe model contains only one linear layer). One epoch is enough for adequate training. And we find that more complicate probe model doesn't have obvious gain.
>
> ### Question
>
> Q1: In this paper, we use linear probe, (The probe model contains only one linear layer). One epoch is enough for adequate training. Training for more epochs doesn't helpful in our problem. And we find that more complicate probe models don't have obvious gain.
>
> Q2: We disable the attention layers by replacing the attention layers in the Transformer with identity maps, whose outputs are equal to their inputs. We add this content in the paper.

---

### Official Review · Reviewer_F4jH · 2023-11-08

**Soundness:** 2 fair
**Presentation:** 3 good
**Contribution:** 3 good
**Rating:** 5
**Confidence:** 3

**Summary:**

This paper sets out to shed further light on the in-context learning abilities of transformers. The paper studies. The authors consider a synthetic pattern-matching tasks called Shapes3D, in which the transformer is tasked with predicting the value of a certain factor that changes in images of 3D shapes (e.g. shape, color, background, etc.). The task can be learned by in-context examples of images of shapes and the corresponding value of the relevant factor (which is unknown). The authors attempt to distinguish between what is learned through the in-context component and what is learned through the in-weights component by analyzing the predictive power of their respective representations. The authors find that they play complementary roles and correlations are elucidated that reveal what is necessary for strong in-context learning capabilities.

**Strengths:**

- The paper studies an important problem. Understanding what is important to achieve in-context learning is quite clearly a very important topic that can have critical downstream applications in understanding and improving the robustness of models such as language models.
- The paper is presented fairly clearly, but there are some things that I would suggest listed below.
- The concepts that the paper attempts to elucidate are fairly new to my knowledge, compared to prior work and the methodology of looking at representations potentially analyzes concepts that were previously not analyzed, which I greatly appreciate. This approach to probing could be  a useful technique to study in-context learning (although probing itself is not new).
- The theory is useful to have a construction that achieves the desired results. However, it is limited to linear attention, but I think this is fine given the precedents.

**Weaknesses:**

Despite these strengths, I have some reservations about the inferences that can be drawn from the study.

- A fundamental (somewhat unstated) assumption is that the in-context comp and in-weights comp are decomposed via this probing analysis and thus we can see the impact of both of them. However, the in-context component is clearly also impacted by concepts learned in the weights. It also already seems obvious that one would need to be able to achieve a high in-context comp score in order to achieve high in-context accuracy. For example, I fail to see what we learn regarding the second major claim ‘in-weights comp plays a crucial role in learning the in-context comp’. I am confused about (1) why this is surprising (2) if it is surprising, which of the experiments justifies this. I could see 6A potentially supporting this, but there are other factors at play. In 5, one trains from scratch so clearly there is some ‘warm up’ that is needed to get in-context learning ability.
- One of the other fundamental pieces of the results is training on D_fix and then evaluating on D_rnd or some variation of this. I am concerned that the conclusions that are drawn from these experiments are not necessarily results of in-context learning properties but are just generally issues related to distribution shift. After all, the transformer is not trained on D_rnd. The trend of the results is the following: when we use D_rnd in training, the results get better when evaluating on D_rnd (Fig 5, 6). Is this somehow unexpected?
- Given this, perhaps the most surprising result is that the in-weights scores on 5A with D_rnd are poor and 6A does not improve to what is achievable training full on D_rnd, but only the in-context score improves. I’m not sure what to make of this though. For instance, this could be explained by saying that it just may not be important to form a representation that is predictive of the factor values without the context for this particular task since the task can be inferred from the context. Perhaps the authors can shed some light on that.

**Questions:**

In addition to some questions, I also suggest some minor changes that I think would improve the paper.


- It would be helpful to more formally define the in-context and in-weights components earlier in the paper. Throughout the intro and early parts, it is clear what th paper wants to do, but it’s difficult to discern what quantifiably one wants to measure since ‘in-weights components’ and ‘in-context components’ are sort of nebulous terms until one reaches section 2.3. Honestly, I don’t know the best way of presenting it, but I just want to raise this as a potential issue for future readers in case the authors have alternatives.
- when you compute the in-context comp score, are you using the very last element of the sequence at L?
- when producing the probes, are you doing a train and test split and are we looking at the test accuracy. How is this split being done? Are we looking at the test score when we look at the plots?
- FYI I don’t think E is defined before being used in Section 2.3.
- Fig 6A should be D_fix->rnd not D_rnd->fix I think.
- Fig 6A why is it D_rnd^fix => D_fix and not D_rnd^fix => D_rnd? Isn’t it more relevant to understand the in-context score on D_rnd?
- In, D_rnd^fix, do you use twice the amount of data as D_rnd? That is, does each epoch contain twice the data or are they both halved so the total is the same?
- I don’t really understand the last paragraph that starts with “Comparison with previous work in analyzing the in-context learning mechanism.”
- It would be helpful to compare 6A directly with 5B since they are plotting the same kind of results. It seems that in both you get good performance starting at roughly the same epochs. This I think supports my question about how much of this is just attributed to distribution shift: once you start training on the same data as you test on, performance improves.
- Some typos: “copy-past” “fead forward” “How transformer solve this problem?” “analysis is to analysis”

---

> ### Author Response · Authors · 2023-11-17
>
> Thank you for your valuable comments and suggestions.  Please find below our responses to your concerns and the changes we have made to address them.
>
> ### Weakness 1.
> **Point: A fundamental (somewhat unstated) assumption is that the in-context comp and in-weights comp are decomposed via this probing analysis. However, the in-context component is clearly also impacted by concepts learned in the weights.**
> In the revised version, we provide a formal definition of the problem along with the assumptions in Section 2.2.
>
> **“Point:It also already seems obvious that one would need to be able to achieve a high in-context comp score in order to achieve high in-context accuracy.**
> I understand the reviewer's logic in associating "in-context comp" and "in-context accuracy" with in-context learning due to their terminology and their relation to in-context examples. However, two points might have been overlooked. 1) The existence of in-context comp isn't natural; it's a result of our effort to decompose it from the representation. 2) Furthermore, our findings suggest a portion of the representation is directly tied to performance. This connection hasn't been explored in previous work, and this discovery suggests that enhancing representation learning may be a fundamental approach to improve in-context learning.
>
>
> **Point: I fail to see what we learn regarding the second major claim ‘in-weights comp plays a crucial role in learning the in-context comp’**
> 1) Regarding the statement, "I could see 6A potentially supporting this, but there are other factors at play. In 5, one trains from scratch so clearly there is some ‘warm up’ that is needed to get in-context learning ability." We've supplemented this with a new experiment (Figure 9A in Appendix F), where we train D_fix for a brief period and still observe a significant improvement.
> 2) The use of both D_{fix} and D_{rnd} leads to considerably better results compared to utilizing D_{rnd} alone.
>
> ### Weakness 2.
>  We agree with you that there is a distribution shift. However, we disgree with your logic that ``Distribution shift->drop of performance".
> The reason is that
> 1) There are no drops of the performance when we swith from D_rnd to D_fix, although there are also "distribution shift" exsits.
> 2) Using D_{rnd and fix} can improve the results on D_rnd compared with only trained on D_rnd. D_{rnd and fix} also introduce the distribution shift.
> Our logic is that `` Distribution shift -> unaffected in-weights component and affected in-context component -> drop of performance".
> Our experiment reveal that training on D_fix and then evaluate on D_rnd, only the learning of the in-context component cannot be transfered. And due to this factor, the performance drops.
>
> ### Weakness 3.
> **Point: Given this, perhaps the most surprising result is that the in-weights score on 5A with D_rnd are poor and 6A does not improve to what is achievable training full on D_rnd, but only the in-context score improves. I’m not sure what to make of this though.**
>
> Here, we give possible reasons for this obervation:
>
> 1) We don't have direct supervision signal for improving in-weights and in-context component.
>
> 2) Under D_rnd setting, the model can achieve smaller loss by just improving the in-context component. Therefore, the model don't have motivation to improve the in-weights component and it even sacrifice in-weights component to obtain better in-context component.
>
> ### Question
>
> Q1 We rewrite the paper and give the formal definition in Section 2.2
>
> Q2 Yes, we use the last element of the sequence at L
>
> Q3 Yes, we look at the test score. We add a description for the dataset split of probe model in Appendix A.4.
>
> Q4, we rewrite the paper to fix the problem.
>
> Q5, Yes, it is a typo. It is D_rnd->D_fix
>
> Q6, It it a typo. Actually it is D_rnd^fix => D_rnd. Thank you for your pointing out. we have fixed it.
>
> Q7，We use same amout of data. We add a description for the dataset split of probe model in Appendix A.4.
>
> Q8, this sentence is related to the proof details of the proposition. I move it to the Appendix for the audiences who have interesting in the proof.
>
> Q9, We give a directly comparison about the some results on Figure 9A (Appendix). Thanks for your advise, we will rethink it and try to find a better way for representation.
>
> Q10, Thank you for your point out. We have fixed it in the paper.

---

### Author Response · Authors · 2023-11-17
**A new revision of paper**

Thanks for AC and reviewers spending time in handling this paper. We thank all the reviewers for the positive feedbacks and constructive comments, as well as their the encouraging words: The paper studies an **important problem**. The concepts that the paper attempts to elucidate are fairly **new to my knowledge** (Reviewer F4jH). The theory is useful to have a construction that achieves the desired results (Reviewer F4jH).  They develop a scoring system for the in-context and in-weight components, **a novel and logical idea** (Reviewer EdZW). the approach of probing in-context and in-weights representation is definitely an **interesting** and the authors take a look at it from representation learning perspective (Reviewer MkMt) and The provided perspective that analyzes in-context learning is interesting and may **inspire follow-up research** (Reviwer gm8Y).


Reviewer EdZW and Reviewer gm8Y concern that current form cannot give a clear definition of the problem, which hurt their understanding.  Reviewer MkMt and Reviewer gm8Y concern whether conclusion of the synthetic tasks can be extended to NLP task. Reviewer EdZW and Reviewer F4jH concerns about the details of probe models. Reviewer MkMt concerns the readability of the proof.

We give the following revision to our papers in order to solve the concerns that raised by reviewers.

1) We rewrite the section 2 to make the the problem and definition more clear. **We high recommend you to read this part.** In the revision, we give a formal definition about in-weights and in-context components, a decomposition of the problem, and an analysis about why we consider these different settings.


2) We add Section C in Appendix to discuss the potential of our work in practice. We summary some prior works that give evidence for both in-weights and in-context components used in practice. And we give a experiment on a new NLP dataset to demonstrate that conclusions obtained in the synthetic tasks can be extended to NLP task.

3) We rewrite the proof to make it more readable. We give a extra paragraph to summary the notations and more explainations/intuitions in the proof.

4) We add more results in Section F in Appendix. These results include the probing results on different layers and D_fix->D_rnd settings with fewer epochs training on D_fix.

---

### Author Response · Authors · 2023-11-23

Thank you for all of your constructive comments and suggestions. Please let us know as soon as possible if you have any further questions or concerns, since the discussion stage is coming to an end soon.

---

### Meta-Review · Area_Chair_Dpq6 · 2023-12-05

**Metareview:**

This paper provides further understandings of in-context learning (ICL) in transformers via the lens of representation learning. The paper constructs ICL tasks from the Shapes3D dataset involving both a fixed representation mapping (to be learned "in weights") and a changing function from representation to label (to be learned in context). The paper then designs experiments to compare the in weights component and the in context component of trained transformers, and study the impact of training data distribution (including the order of appearance of different training distributions). The paper also provides some theoretical results.

While the reviewers generally agree with the importance of the topic and that the setup could be promising, none of them are supportive of the paper, due to serious concerns about the soundness of the conclusions as well as the clarity of the presentation. Unfortunately, given these concerns, the paper is not ready for publication at ICLR in its current stage.

I encourage the authors to do a thorough polishing of the presentation (such as more formal definition of the high-level concepts, simpler math notation, clearer description of the methodologies) and take into account the reviewers suggestions, which would significantly improve the paper.

**Justification For Why Not Higher Score:**

There are serious concerns about the soundness of the conclusions given the current experiments, as well as the clarity of the setups and methodologies.

**Justification For Why Not Lower Score:**

N/A

---

### Decision · Program_Chairs · 2024-01-16

Reject